# Topologically faithful image segmentation via induced matching of persistence barcodes

## Abstract

Image segmentation is a largely researched field where neural networks find vast applications in many facets of technology. Some of the most popular approaches to train segmentation networks employ loss functions optimizing pixel-overlap, an objective that is insufficient for many segmentation tasks. In recent years, their limitations fueled a growing interest in topology-aware methods, which aim to recover the correct topology of the segmented structures. However, so far, none of the existing approaches achieve a spatially correct matching between the topological features of ground truth and prediction.

In this work, we propose the first topologically and feature-wise accurate metric and loss function for supervised image segmentation, which we term *TopoMatch*. We show how induced matchings guarantee the spatially correct matching between barcodes in a segmentation setting. Furthermore, we propose an efficient algorithm to compute TopoMatch for images. We show that TopoMatch is an interpretable metric to evaluate the topological correctness of segmentations, which is more sensitive than the well-established Betti number error. Moreover, the differentiability of the *TopoMatch loss* enables its use as a loss function. It improves the topological performance of segmentation networks across six diverse datasets while preserving the volumetric performance.

## 1 Introduction

Topology studies properties of shapes that are related to their connectivity and that remain unchanged under deformations, translations, and twisting. Some topological concepts, such as *cubical complexes*, *homology* and *Betti numbers*, form interpretable descriptions of shapes in space that can be efficiently computed. Naturally, the topology of physical structures is highly relevant in machine learning tasks, where the preservation of its connectivity is crucial, a prominent example being image segmentation. Recently, a number of methods have been proposed to improve topology preservation in image segmentation for a wide range of applications. However, none of the existing concepts take the spatial location of the topological features (e.g. *connected components* or *cycles*) within their respective image into account. Evidently, spatial correspondence of these features is a critical property of segmentations, see Fig. 1.

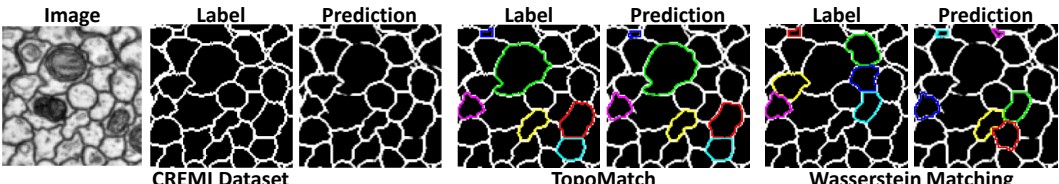

Figure 1: Motivation – comparison of our *TopoMatch* and *Wasserstein matching* (Hu et al. (2019)). We match cycles between label and prediction for a CREMI image and denote matched pairs in the same color. We visualize only six (randomly selected out of the total 23 matches for both methods) matched pairs for presentation clarity. Note that TopoMatch always matches spatially correctly while the Wasserstein matching gets most matches wrong.

**Our contribution** In this work, we introduce a rigorous framework for *faithfully* quantifying the preservation of topological properties in the context of image segmentation, see Fig. 2. Our method

builds on the concept of *induced matchings* between *persistence barcodes* from algebraic topology, introduced by Bauer & Lesnick (2015). The introduction of these matching to a machine learning setting allows us to precisely formulate spatial correspondences between topological features of two grayscale images. We achieve this by embedding both images into a common *comparison image*. Put in simple terms, our central contribution is an efficient, differentiable solution for localized topological error finding, which serves as:

- a **topological loss** to train segmentation networks, which guarantees to correctly, in a spatial sense, emphasize and penalize the topological structures during training (see Sec 3.2);

- an **interpretable topological quality metric** for image segmentation, which is not only sensitive to the number of topological features but also to their location within the respective images (see Sec. 3.3).

Experimentally, our TopoMatch construction proves to be an effective loss function, leading to vastly improved topology across six diverse datasets.

## 1.1 RELATED WORK

**Algebraic stability of persistence** Several proofs for the stability of persistence have been proposed in the literature. In 2005, Cohen-Steiner et al. (2005) established a first stability result for *persistent homology* of real-valued functions. The result states that the map sending a function to the *barcode* of its sublevel sets is 1-Lipschitz with respect to suitable metrics. In 2008 this result was generalized by Chazal et al. (2009b) and formulated in purely algebraic terms, in what is now known as the algebraic stability theorem. It states that the existence of a $\delta$-interleaving (a sort of approximate isomorphism) between two pointwise finite-dimensional persistence modules implies the existence of a $\delta$-matching between their respective barcodes. This theorem provides the justification for the use of persistent homology to study noisy data. In Bauer & Lesnick (2015), the authors present a constructive proof of this theorem, which associates to a given $\delta$-interleaving between persistence modules a specific $\delta$-matching between their barcodes. For this purpose, they introduce the notion of induced matchings, which form the foundation of our proposed TopoMatch framework.

**Topology aware segmentation** Multiple works have highlighted the importance of topologically correct segmentations in various computer vision applications. *Persistent homology* is a popular tool from algebraic topology to address this issue. A key publication by Hu et al. (2019) introduced the *Wasserstein loss* as a variation of the *Wasserstein distance* to improve image segmentation. They match points of dimension 1 in the *persistence diagrams* – an alternative to barcodes as descriptor of persistent homolgy – of ground truth and prediction by minimizing the squared distance of matched points.

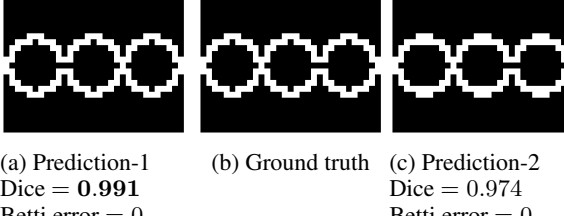

(a) Prediction-1
Dice = **0.991**
Betti error = 0
TopoMatch = 1

(b) Ground truth

(c) Prediction-2
Dice = 0.974
Betti error = 0
TopoMatch = **0**

Figure 2: (a) and (c) show two predictions for ground truth (b). Volumetric metrics, e.g., *Dice* favor (a) over (c), and even Betti number error can not differentiate between (a) and (c) while only TopoMatch detects the spatial error in (a) and favors (c).

However, this matching has a fundamental limitation, in that it cannot guarantee that the matched structures are spatially related in any sense (see Fig. 1 and App. A). Put succinctly, the cycles are matched irrespective of the location within the image, which frequently has an adverse impact during training (see App. F). Clough et al. (2020) follows a similar approach and train without knowing the explicit ground truth segmentation, but only the Betti numbers it ought to have. Persistent homology has also been used by Abousamra et al. (2021) for crowd localization and by Waibel et al. (2022) for reconstructing 3D cell shapes from 2D images.

Other methods incorporate pixel-overlaps of topologically relevant structures. For example, the *clDice* score, introduced by Shit et al. (2021), computes the harmonic mean of the overlap of the predicted skeleton with the ground truth volume and vice versa. Hu & Chen (2021) and Jain et al. (2010) use *homotopy warping* to identify critical pixels and measure the topological difference between grayscale images. Hu et al. (2021) utilizes *discrete Morse theory* (see Delgado-Friedrichs

et al. (2014)) to compare critical topological structures within prediction and ground truth. Wang et al. (2022) incorporate a *marker loss*, which is based on the Dice loss between a predicted marker map and the ground truth marker map, to improve gland segmentations topologically. Generally, these overlap-based approaches are computationally efficient but do not explicitly guarantee the spatial correspondence of the topological features. Other approaches aim at enforcing topologically motivated priors, for example, enforcing connectivity priors Sasaki et al. (2017); Wang & Jiang (2018).Mosinska et al. (2018) applied task-specific pre-trained filters to improve connected components. Zhang & Lui (2022) uses template masks as an input to enforce the diffeomorphism type of a specific shape. Further work by Cheng et al. (2021) jointly models connectivity and features based on iterative feedback learning. Oner et al. (2020) aims to improve the topological performance by enforcing region separation of curvilinear structures.

## 2 BACKGROUND ON ALGEBRAIC TOPOLOGY

We introduce the necessary concepts from algebraic topology in order to describe the construction of induced matchings for grayscale images. For the basic definitions, we refer to the App. L.

### 2.1 GRAYSCALE IMAGES AS FILTERED CUBICAL COMPLEXES

The topology of grayscale images is best captured by *filtered cubical complexes*. In order to filter a *cubical complex* $K$ we consider an *order preserving* function $f \colon K \to \mathbb{R}$. Its **sublevel sets** $D(f)_r := f^{-1}((-\infty, r])$ assemble to the **sublevel filtration** $D(f) = \{D(f)_r\}_{r \in \mathbb{R}}$ of $K$. Since $f$ can only take finitely many values $\{f_1 < \ldots < f_l\}$, the filtered cubical complex $K_*$ given by $K_i = D(f)_{f_i}$ for $i = 1, \ldots, l$, already encodes all the information about the filtration.

For a grayscale image $\boldsymbol{I} \in \mathbb{R}^{m \times n}$ we consider the **cubical grid complex** $K^{m,n}$ consisting of all cubical cells contained in $[1, m] \times [1, n] \subseteq \mathbb{R}^2$. Its **filter function** $f_{\boldsymbol{I}}$ is defined on the vertices of $K^{m,n}$ by the corresponding entry in $\boldsymbol{I}$, and on all higher-dimensional cubes as the maximum value of its vertices. Note that $f_{\boldsymbol{I}}$ is order preserving, so we can associate the sublevel filtration of $f_{\boldsymbol{I}}$ and its corresponding filtered cubical complex to the image $\boldsymbol{I}$ and denote them by $D(\boldsymbol{I})$ and $K_*(\boldsymbol{I})$, respectively. This construction is called the **V-construction** since pixels are treated as vertices in the cubical complex, see Fig. 3b. An alternative, the **T-construction**, considers pixels

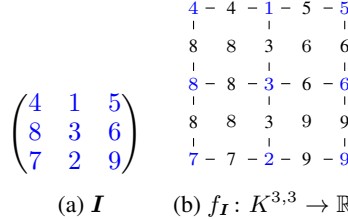

(a) $\boldsymbol{I}$    (b) $f_{\boldsymbol{I}} \colon K^{3,3} \to \mathbb{R}$

Figure 3: (a) shows an image and (b) visualizes the V-construction.

as top-dimensional cells of a 2-dimensional cubical complex (see Heiss & Wagner (2017)). We implemented both, V- and T-construction, in TopoMatch and encode them in the *ValueMap* array inside the *CubicalPersistence* class.

### 2.2 PERSISTENT HOMOLOGY AND ITS BARCODE

Persistent homology considers filtrations of spaces and observes the lifetime of topological features within the filtration in form of *persistence modules*. The basic premise is that features that persist for a long time are significant, whereas features with a short lifetime are likely to be caused by noise.

The **persistent homology** $H_*(f)$ of an order preserving function $f \colon K \to \mathbb{R}$ consists of vector spaces $H_*(f)_r = H_*(D(f)_r)$ and transition maps $H_*(f)_{r,s} \colon H_*(D(f)_r) \to H_*(D(f)_s)$ induced by the inclusions $D(f)_r \hookrightarrow D(f)_s$ for $r \leq s$. Note that $H_*(f)$ is a p.f.d persistence module. By a result of see Crawley-Boevey (2012), any p.f.d. persistence module is isomorphic to a direct sum

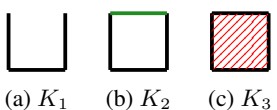

(a) $K_1$    (b) $K_2$    (c) $K_3$

Figure 4: A filtered cubical complex with varying homology in degree 1. Adding the green 1-cell in (b) creates homology and adding the red 2-cell in (c) turns homology trivial. Together they form a persistence pair.

of interval modules $M \cong \bigoplus_{I \in \mathcal{B}(M)} C(I)$. Here, $\mathcal{B}(M)$ denotes the **barcode** of $M$, which is given by a *multiset* of intervals. For a grayscale image matrix $\boldsymbol{I} \in \mathbb{R}^{m \times n}$ with associated filter function $f_{\boldsymbol{I}} \colon K^{m,n} \to \mathbb{R}$, we will refer to the persistent homology of $f_{\boldsymbol{I}}$ as the persistent homology of the image $\boldsymbol{I}$ and denote it by $H_*(\boldsymbol{I})$. Its associated barcode will be denoted by $\mathcal{B}(\boldsymbol{I})$. Note that the persistent homology is **continuous from above**: all intervals in the barcode are of the form $[s, t)$.

In order to compute the barcode $\mathcal{B}(\boldsymbol{I})$, we make use of the *reduction algorithm* described in Edelsbrunner et al. (2008). It starts by sorting the cells of the associated filtered cubical complex $K_*(\boldsymbol{I})$ to obtain a **compatible** ordering $c_1, \ldots, c_l$: the cells in $K_i$ preceed the cells in $K \setminus K_i$, and the faces of a cell preceed the cell. This ordering induces a *cell-wise refinement* $L_*(\boldsymbol{I})$ of $K_*(\boldsymbol{I})$, which we encode in the *IndexMap* array inside the CubicalPersistence class. The algoritm then performs a variant of Gaussian elimination on the *boundary matrix* of $K$, with rows and columns indexed by the cells in the compatible ordering. Adding a $d$-cell $c_k$ to the complex will either create new homology in degree $d$ or turn homology classes in degree $d-1$ trivial (see Figure 4). In the latter case, assuming that the class that becomes trivial when adding $c_k$ has been created by cell $c_j$ with $j < k$, we pair the cells $c_j$ and $c_k$. This way we partition the set of cells into **persistence pairs** and singletons. Each pair $(c_j, c_k)$, satisfying $f_{\boldsymbol{I}}(c_j) < f_{\boldsymbol{I}}(c_k)$, gives rise to a **finite interval** $[f_{\boldsymbol{I}}(c_j), f_{\boldsymbol{I}}(c_k))$, and each singleton $c_i$ gives rise to an **essential interval** $[f_{\boldsymbol{I}}(c_i), \infty)$ in the barcode of $\boldsymbol{I}$. Note that a finite interval $[f_{\boldsymbol{I}}(c_j), f_{\boldsymbol{I}}(c_k))$ determines a **refined interval** $[j, k)$, and we call the set $\mathcal{B}_{\text{fine}}(\boldsymbol{I})$ of refined intervals the **refined barcode** of $\boldsymbol{I}$. Alternatively, the refined barcode of $\boldsymbol{I}$ can be seen as barcode of the persistent homology of the refined filtration $L_*(\boldsymbol{I})$.

## 2.3 INDUCED MATCHINGS OF PERSISTENCE BARCODES

In order to give a constructive proof for the algebraic stability theorem of persistent homology, the authors of Bauer & Lesnick (2015) introduced the notion of induced matchings, which play a central role in our TopoMatch matching. The following theorem (paraphrased as a special case of the general Theorem 4.2 in Bauer & Lesnick (2015)) is key to the definition of induced matchings:

**Theorem 1** *Let $\Phi \colon M \to N$ be a morphism of p.f.d., staggered persistence modules that are continuous from above. Then there are unique injective maps $\mathcal{B}(\operatorname{im}\Phi) \hookrightarrow \mathcal{B}(M)$ and $\mathcal{B}(\operatorname{im}\Phi) \hookrightarrow \mathcal{B}(N)$, which map an interval $[b, c) \in \mathcal{B}(\operatorname{im}\Phi)$ to an interval $[b, d) \in \mathcal{B}(M)$ with $c \leq d$, and to an interval $[a, c) \in \mathcal{B}(N)$ with $a \leq b$, respectively.*

Note that $\operatorname{im}\Phi$ is a p.f.d. submodule of $N$, and we will refer to its barcode as the **image barcode** of $\Phi$. Obviously, the injections in Theorem 1 determine matchings $\mathcal{B}(M) \xrightarrow{\sigma_M} \mathcal{B}(\operatorname{im}\Phi) \xrightarrow{\sigma_N} \mathcal{B}(N)$. The **induced matching** of $\Phi$ is then given by the composition $\sigma(\Phi) = \sigma_N \circ \sigma_M$.

**Induced matchings of grayscale images** Let $\boldsymbol{I}, \boldsymbol{J} \in \mathbb{R}^{m \times n}$ be matrices describing grayscale images, such that $\boldsymbol{I} \geq \boldsymbol{J}$ (entry-wise). Then the sublevel sets of $\boldsymbol{I}$ form subcomplexes of the sublevel sets of $\boldsymbol{J}$ and the corresponding inclusions $D(\boldsymbol{I})_r \hookrightarrow D(\boldsymbol{J})_r$ are cubical maps. Hence, they induce maps $H_*(\boldsymbol{I})_r \to H_*(\boldsymbol{J})_r$ in homology, which assemble to a persistence map $\Phi(\boldsymbol{I}, \boldsymbol{J}) \colon H_*(\boldsymbol{I}) \to H_*(\boldsymbol{J})$. We will denote the image barcode of $\Phi(\boldsymbol{I}, \boldsymbol{J})$ by $\mathcal{B}(\boldsymbol{I}, \boldsymbol{J})$. Considering the refined filtrations $L_*(\boldsymbol{I}), L_*(\boldsymbol{J})$, we obtain staggered persistence modules resulting in refined barcodes $\mathcal{B}_{\text{fine}}(\boldsymbol{I}), \mathcal{B}_{\text{fine}}(\boldsymbol{J})$. For the computation of the image barcode, we follow the algorithm described in Bauer & Schmahl (2022). It involves the *reduction* of the boundary matrix of $K^{m,n}$ with rows indexed by the ordering $c_1, \ldots, c_l$ in $L_*(\boldsymbol{I})$ and columns indexed by the ordering $d_1, \ldots, d_l$ in $L_*(\boldsymbol{J})$. A pair $(c_i, d_j)$ satisfying $f_{\boldsymbol{I}}(c_i) < f_{\boldsymbol{J}}(d_j)$, obtained by the means of this reduction, then gives rise to an **image persistence pair** $(c_i, d_j)$, which corresponds to the finite interval $[f_{\boldsymbol{I}}(c_i), f_{\boldsymbol{J}}(d_j)) \in \mathcal{B}(\boldsymbol{I}, \boldsymbol{J})$. By matching refined intervals with the image persistence pairs according to Theorem 1, we obtain a matching $\sigma_{\text{fine}} \colon \mathcal{B}_{\text{fine}}(\boldsymbol{I}) \to \mathcal{B}_{\text{fine}}(\boldsymbol{J})$ between the refined barcodes, which determines the **induced matching** $\sigma(\boldsymbol{I}, \boldsymbol{J}) \colon \mathcal{B}(\boldsymbol{I}) \to \mathcal{B}(\boldsymbol{J})$ by replacing refined intervals with the corresponding finite interval.

$$\begin{pmatrix} 0 & 1 & 2 \\ 7 & 39 & 3 \\ 6 & 5 & 4 \end{pmatrix}$$
(a) $\boldsymbol{J}_1$

(b) $\sigma(\boldsymbol{I}, \boldsymbol{J}_1)$

$$\begin{pmatrix} 20 & 27 & 26 \\ 21 & 49 & 25 \\ 22 & 23 & 24 \end{pmatrix}$$
(c) $\boldsymbol{I}$

(d) $\sigma(\boldsymbol{I}, \boldsymbol{J}_2)$

$$\begin{pmatrix} 0 & 1 & 2 \\ 7 & 19 & 3 \\ 6 & 5 & 4 \end{pmatrix}$$
(e) $\boldsymbol{J}_2$

Figure 5: (a), (c) and (e) show images which satisfy $\boldsymbol{I} \geq \boldsymbol{J}_1, \boldsymbol{J}_2$. (b) and (d) visualize the induced matchings. Red bars correspond to the barcode of $\boldsymbol{I}$, green bars to the barcodes of $\boldsymbol{J}_1, \boldsymbol{J}_2$ and grey bars to the image barcodes $\mathcal{B}(\boldsymbol{I}, \boldsymbol{J}_1), \mathcal{B}(\boldsymbol{I}, \boldsymbol{J}_2)$. The shaded gray area highlights matched intervals according to their endpoints.

In this work, we augment this induced matching by additionally considering **reverse persistence pairs**, i.e., pairs $(c_i, d_j)$ obtained by the reduction, with $f_{\boldsymbol{I}}(c_i) \geq f_{\boldsymbol{J}}(d_j)$ (see Figure 5d). When this is the case, we also match the corresponding intervals in $\mathcal{B}_{\text{fine}}(\boldsymbol{I})$ and $\mathcal{B}_{\text{fine}}(\boldsymbol{J})$. Note that this is a slight variation of the induced matching defined in Bauer & Lesnick (2015). This extension satisfies similar properties and is a natural adaptation to our computational context.

## 3 TOPOMATCH – FROM ALGEBRAIC TOPOLOGY TO IMAGE SEGMENTATION

In general, the structure of interest in segmentation tasks is given by the foreground. Therefore, we consider *superlevel filtrations* instead of sublevel filtrations in applications. For simplicity, we stick to sublevel filtrations to describe the theoretical background. Throughout this section, we denote by $\boldsymbol{L} \in [0, 1]^{m \times n}$ a **likelihood map** predicted by a deep neural network, by $\boldsymbol{P} \in \{0, 1\}^{m \times n}$ the binarized **prediction** of $\boldsymbol{L}$, and by $\boldsymbol{G} \in \{0, 1\}^{m \times n}$ the **ground truth** segmentation.

### 3.1 MATCHING BY COMPARISON IN AMBIENT SPACE

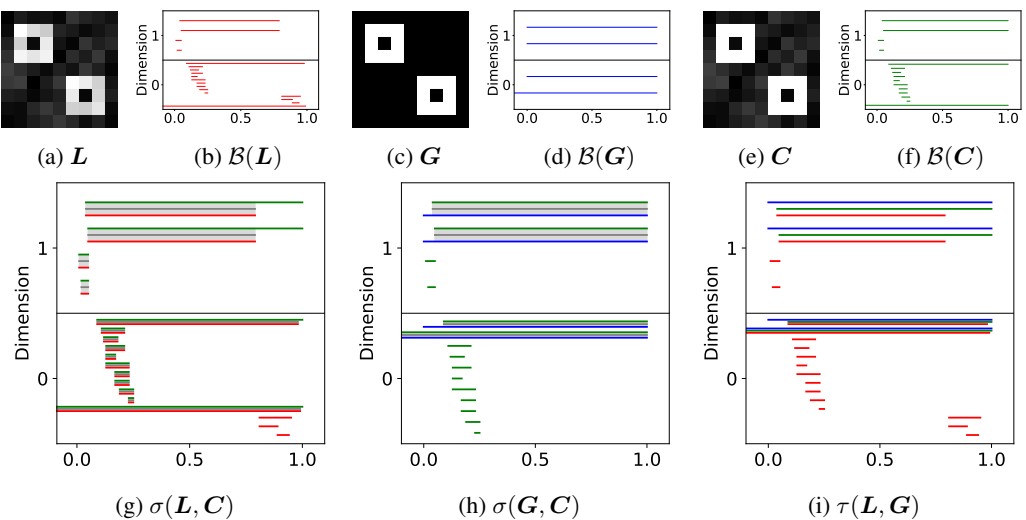

Figure 6: An exemplary construction of the TopoMatch matching. (a)–(f) show a likelihood map $\boldsymbol{L}$, a ground truth $\boldsymbol{G}$, the comparison image $\boldsymbol{C}$ and their barcodes. (g) and (h) show the induced matchings between individual barcodes (matchings indicated in grey), and (i) shows the resulting TopoMatch matching between $\mathcal{B}(\boldsymbol{L})$ and $\mathcal{B}(\boldsymbol{G})$, which matches a red interval to a blue interval if there is a green interval in between. We use this matching to define our loss and metric.

In order to visualize that two objects in two different images are at the same location, we can simply move one image ontop of the other one and observe that the locations of the objects now agree. Thereby, we are constructing a common ambient space for both images which allows us to identify locations. Following this idea, in order to find a matching between $\mathcal{B}(\boldsymbol{L})$ and $\mathcal{B}(\boldsymbol{G})$ that takes the location of represented topological features into account, we are looking for a common *ambient filtration* of $K^{m,n}$, which is

(a) big enough to contain the sublevel filtrations of $\boldsymbol{L}$ and $\boldsymbol{G}$;

(b) fine enough to capture the topologies of $\boldsymbol{L}$ and $\boldsymbol{G}$.

Here, (a) guarantees that we can compute induced matchings of the respective inclusions and (b) guarantees that the identification of features by the induced matchings are non-trivial (discriminative). The most natural candidate which comes into mind is given by the union $D(\boldsymbol{L})_r \cup D(\boldsymbol{G})_r$ of sublevel sets. Therefore, we introduce the comparison image $\boldsymbol{C} = \min(\boldsymbol{L}, \boldsymbol{G})$ (entry-wise minimum) and observe that $D(\boldsymbol{C})_r = D(\boldsymbol{L})_r \cup D(\boldsymbol{G})_r$. By construction, we have $\boldsymbol{C} \leq \boldsymbol{L}, \boldsymbol{G}$ and obtain induced matchings $\sigma(\boldsymbol{L}, \boldsymbol{C}) \colon \mathcal{B}(\boldsymbol{L}) \to \mathcal{B}(\boldsymbol{C})$ and $\sigma(\boldsymbol{G}, \boldsymbol{C}) \colon \mathcal{B}(\boldsymbol{G}) \to \mathcal{B}(\boldsymbol{C})$ (see Sec. 2.3). The **TopoMatch matching** $\tau(\boldsymbol{L}, \boldsymbol{G}) \colon \mathcal{B}(\boldsymbol{L}) \to \mathcal{B}(\boldsymbol{G})$ is then given by the composition

$$\tau(\boldsymbol{L}, \boldsymbol{G}) = \sigma(\boldsymbol{G}, \boldsymbol{C})^{-1} \circ \sigma(\boldsymbol{L}, \boldsymbol{C}).$$

Working with superlevel sets yields an analogous construction. In the superlevel-setting we choose $C = \max(L, G)$ as the comparison image to guarantee that each superlevel set of the comparison image is the union of the corresponding superlevel sets of ground truth and likelihood map.

### 3.2 TOPOMATCH DEFINES A TOPOLOGICAL LOSS FUNCTION FOR IMAGE SEGMENTATION

We denote by $\overline{\mathbb{R}}$ the **extended real line** $\mathbb{R} \cup \{-\infty, \infty\}$. A barcode $\mathcal{B}$ consisting of intervals $[a, b)$ can then equivalently be seen as a multiset $\mathrm{Dgm}(\mathcal{B})$ of points $(a, b) \in \overline{\mathbb{R}}^2$ which lie above the diagonal $\Delta = \{(x, x) \mid x \in \mathbb{R}\}$. Furthermore, we add all the points on the diagonal $\Delta$ with infinite multiplicity to $\mathrm{Dgm}(\mathcal{B})$ and thus define the **persistence diagram** of $\mathcal{B}$. A matching $\sigma \colon \mathcal{B}_1 \to \mathcal{B}_2$ between barcodes then corresponds to a bijection $\sigma \colon \mathrm{Dgm}(\mathcal{B}_1) \to \mathrm{Dgm}(\mathcal{B}_2)$ between persistence diagrams, by mapping unmatched points $(a, b)$ to their closest point $((a + b)/2, (a + b)/2)$ on the diagonal $\Delta$. We use these perspectives interchangeably (see Fig. 20). For simplicity, we denote by $\mathrm{Dgm}(I)$ the persistence diagram associated to the barcode of a grayscale image $I$.

Persistent homology is stable, see Chazal et al. (2009a), i.e., there exist metrics on the set of persistence diagrams for which slight variations in the input result in small variations of the corresponding persistence diagram. Therefore, it is natural to require $\mathrm{Dgm}(L)$ to be similar to $\mathrm{Dgm}(G)$. A frequently used metric to measure the difference between persistence diagrams is the Wasserstein distance (see Cohen-Steiner et al. (2010)), and it has been adapted in Hu et al. (2019) to train segmentation networks. Because of the shortcomings described in Fig. 1,8c,19b and App. A,F, we propose to replace the Wasserstein matching $\gamma_*$ by the TopoMatch matching $\tau(L, G)$ and define the **TopoMatch loss**

$$L_{\mathrm{TM}}(L, G) = \sum_{q \in \mathrm{Dgm}(L)} \|q - \tau(L, G)(q)\|_2^2.$$

Since the values in $L$ and $G$ are contained in $[0, 1]$, we replace the essential intervals $[a, \infty)$ with the finite interval $[a, 1]$, to obtain a well-defined expression. To efficiently train segmentation networks, we combine our TopoMatch loss with a standard volumetric loss, specifically, the *Dice Loss*, to

$$L_{train} = \alpha L_{\mathrm{TM}}(L, G) + L_{\mathrm{dice}}(L, G).$$

**Gradient of TopoMatch loss** Note that we can see $L = L(I, \omega)$ as a function that assigns the predicted likelihood map to an image $I \in \mathbb{R}^{m \times n}$ and the segmentation network parameters $\omega \in \mathbb{R}^l$. A point $q = (q_1, q_2) \in \mathrm{Dgm}(L)$ describes a topological feature that is born by adding pixel $b(q)$ (**birth** of $q$) and killed by adding pixel $d(q)$ (**death** of $q$) to the filtration. The coordinates of $q$ are then determined by their values $q_1 = L_{d(q)}$ and $q_2 = L_{b(q)}$. Assuming that the TopoMatch matching is constant in a sufficiently small neighborhood around the given predicted likelihood map $L$, the TopoMatch loss is differentiable in $\omega$ and the chain rule yields the gradient

$$\nabla_\omega L_{\mathrm{TM}}(L, G) = \sum_{q \in \mathrm{Dgm}(L)} 2(q_1 - \tau(L, G)(q)_1) \frac{\partial L_{d(q)}}{\partial \omega} + 2(q_2 - \tau(L, G)(q)_2) \frac{\partial L_{b(q)}}{\partial \omega}.$$

Note that likelihood maps for which this assumption is not satisfied may exist. But this requires $L$ to have at least two entries with the exact same value, and the set of such likelihood maps has *Lebesgue measure* zero. Therefore, the gradient is well-defined almost everywhere, and in the edge cases, we consider it as a sub-gradient, which still reduces the loss and has a positive effect on the topology of the segmentation.

**Physical meaning of the gradient** To understand the effect of the TopoMatch gradient during training, consider the example in Fig. 7. Let $x, y \in \mathrm{Dgm}(L)$ denote the points corresponding to the yellow and blue cycle in (c), respectively. (b) shows that $x$ is matched and $y$ is unmatched. Since, all points in $\mathrm{Dgm}(G)$ are of the form (0,1), TopoMatch maps $x$ to (0, 1) and $y$ to its closest point $(\frac{y_1+y_2}{2}, \frac{y_1+y_2}{2})$ on

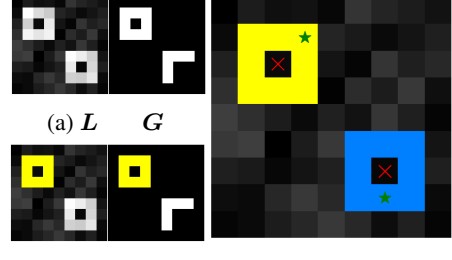

(a) $L$     $G$

(b) TopoMatch     (c) 1-cycles in $L$

Figure 7: (a) $L$ shows a Topological error (bottom right). (b) Matched cycles in TopoMatch are shown in yellow. (c) For both cycles in $L$, the birth ($b(q)$) and death pixels ($d(q)$) are marked with $\star$ and $\times$, respectively.

the diagonal $\Delta$. Therefore, the gradient will enforce the segmentation network to move $x$ closer to $(0, 1)$ (i.e., decrease $x_1 = \boldsymbol{L}_{d(x)}$ and increase $x_2 = \boldsymbol{L}_{b(x)}$) and $y$ closer to $(\frac{y_1+y_2}{2}, \frac{y_1+y_2}{2})$ (i.e., increase $y_1 = \boldsymbol{L}_{d(y)}$ and decrease $y_2 = \boldsymbol{L}_{b(y)}$). This results in an amplification of the local contrast between $\star$ and $\times$ of the yellow cycle and a reduction of the local contrast between $\star$ and $\times$ of the blue cycle, which improves the topological performance of the segmentation.

Summarized, we can say that matched features get emphasized, and unmatched features get suppressed during training, which highlights the importance of finding a spatially correct matching (see App. F for further discussion).

### 3.3 TOPOMATCH LOSS AS A TOPOLOGICAL METRIC FOR IMAGE SEGMENTATION

**Betti number error** The Betti number error $\beta_{\text{err}}$ (see App. K) compares the topological complexity of the binarized prediction $\boldsymbol{P}$ and the ground truth $\boldsymbol{G}$. However, it is limited as it only compares the number of topological features in both images, while ignoring their spatial correspondence (see Fig. 8). In terms of persistence diagrams, the Betti number error can be expressed by considering a maximal matching $\beta \colon \text{Dgm}(\boldsymbol{P}) \to \text{Dgm}(\boldsymbol{G})$, e.g., the Wasserstein matching (see App. F), and counting the number of unmatched points:

$$\beta_{\text{err}}(\boldsymbol{P}, \boldsymbol{G}) = \# \ker(\beta) + \# \text{coker}(\beta).$$

Here, for a matching $\sigma$ we denote by $\ker(\sigma)$ the multiset of unmatched points in the domain of $\sigma$ and by $\text{coker}(\sigma)$ the multiset of unmatched points in the codomain of $\sigma$ (see App. L.5).

**TopoMatch error** The TopoMatch loss $L_{\text{TM}}(\boldsymbol{P}, \boldsymbol{G})$ can be seen as a refinement of the Betti number error, which also takes the location of the features within their respective images into account (see Fig. 8). Since the entries of $\boldsymbol{P}$ and $\boldsymbol{G}$ take values in $\{0, 1\}$, the only point appearing in their persistence diagrams is $(0, 1)$ and its multiplicity coincides with the number of features in the respective image. Observe that an unmatched point contributes with $(0 - \frac{1}{2})^2 + (1 - \frac{1}{2})^2 = \frac{1}{2}$ to $L_{\text{TM}}(\boldsymbol{P}, \boldsymbol{G})$ and a matched pair of points contributes with $0$. Hence, the TopoMatch loss takes values in $\frac{1}{2}\mathbb{N}_0$ and is given by half the number of unmatched features in both $\boldsymbol{P}$ and $\boldsymbol{G}$, i.e.

$$L_{\text{TM}}(\boldsymbol{P}, \boldsymbol{G}) = \frac{1}{2}\big(\# \ker(\tau(\boldsymbol{P}, \boldsymbol{G})) + \# \text{coker}(\tau(\boldsymbol{P}, \boldsymbol{G}))\big).$$

(a) $\beta_{\text{err}}(\boldsymbol{P}, \boldsymbol{G}) = 0$      (b) $L_{\text{TM}}(\boldsymbol{P}, \boldsymbol{G}) = 2$      (c) $L_{\text{W}}(\boldsymbol{P}, \boldsymbol{G}) = 0$

Figure 8: Illustration of the advantages of our TopoMatch error over the Betti number error. (a) shows a prediction $\boldsymbol{P}$ (left), ground truth $\boldsymbol{G}$ (right) and the corresponding Betti number error. (b) shows the TopoMatch matching in dim-1 (no features are matched) with its corresponding loss and (c) shows the Wasserstein matching in dim-1 (same color indicates a matching) with its corresponding loss. Note that both Betti number error and Wasserstein loss fail to represent the spatial error, while TopoMatch correctly does not match any cycles resulting in a loss of 2.

## 4 EXPERIMENTATION

**Datasets** We employ a set of six datasets with diverse topological features for our validation experimentation. Two datasets, the Massachusetts roads dataset, and the CREMI neuron segmentation dataset, exhibit frequently connected curvilinear, network-like structures, which form a large number of cycles in the foreground. The C.elegans infection live/dead image dataset (Elegans) from the Broad Bioimage Benchmark Collection Ljosa et al. (2012) and our synthetic, modified MNIST dataset LeCun (1998) (synMnist) consist of a balanced number of dimension-0 and dimension-1 features. And third, the colon cancer cell dataset (Colon) from the Broad Bioimage Benchmark Collection Carpenter et al. (2006); Ljosa et al. (2012) and the Massachusetts buildings dataset (Buildings) Mnih (2013) have "blob-like" foreground structures. They contain very few dimension-1 features but every instance of a cell or building forms a dimension-0 feature.

Table 1: Main results for TopoMatch and three baselines on six datasets. Green columns indicate the topological metrics. Bold numbers highlight the best performance for a given dataset if it is significant (i.e. the second best performance is not within std/8). We find that TopoMatch improves the segmentations in all topological metrics for all datasets. We further observe a constantly high performance in volumetric metrics. ↑ indicates higher value wins and ↓ the opposite.

| | Loss | Dice ↑ | clDice ↑ | Acc. ↑ | T.M.↓ | T.M.-0↓ | T.M.-1↓ | Betti ↓ | Betti-0 ↓ | Betti-1 ↓ |
|---|---|---|---|---|---|---|---|---|---|---|
| **CREMI** | Dice | 0.894 | 0.939 | 0.959 | 74.82 | 19.84 | 54.98 | 114.12 | 39.12 | 75.00 |
| | clDice | 0.879 | **0.944** | 0.952 | 73.52 | 17.18 | 56.34 | 103.92 | 33.64 | 70.28 |
| | Hu et al. | 0.888 | 0.935 | 0.957 | 81.24 | 22.12 | 59.12 | 118.16 | 43.68 | 74.48 |
| | Ours | 0.893 | 0.941 | 0.959 | **64.90** | **15.50** | **49.40** | **79.16** | **30.36** | **48.80** |
| **Roads** | Dice | 0.663 | 0.698 | 0.974 | 58.90 | 43.52 | 15.38 | 113.96 | 86.54 | 27.42 |
| | clDice | 0.668 | 0.704 | 0.975 | 65.50 | 51.04 | 14.46 | 125.83 | 101.67 | 24.17 |
| | Hu et al. | 0.674 | 0.712 | 0.974 | 50.50 | 36.52 | 13.98 | 95.83 | 72.54 | 23.29 |
| | Ours | 0.663 | 0.713 | 0.972 | **41.50** | **28.15** | 13.35 | **75.08** | **55.79** | **19.29** |
| **synMnist** | Dice | 0.871 | 0.907 | 0.962 | 1.849 | 0.979 | 0.870 | 2.590 | 1.674 | 0.916 |
| | clDice | 0.875 | 0.921 | 0.963 | 1.270 | 0.436 | 0.834 | 1.640 | 0.700 | 0.940 |
| | Hu et al. | 0.866 | 0.915 | 0.960 | 1.425 | 0.502 | 0.923 | 1.802 | 0.764 | 1.038 |
| | Ours | 0.849 | 0.915 | 0.954 | **1.140** | **0.265** | 0.875 | **1.348** | **0.426** | 0.922 |
| **Elegans** | Dice | 0.922 | 0.959 | 0.984 | 2.05 | 1.30 | 0.750 | 2.60 | 1.40 | 1.20 |
| | clDice | 0.917 | **0.964** | 0.982 | 1.95 | 1.10 | 0.850 | 2.20 | 1.20 | 1.00 |
| | Hu et al. | 0.921 | 0.959 | 0.984 | 2.15 | 1.42 | 0.725 | 2.50 | 1.35 | 1.15 |
| | Ours | 0.919 | 0.960 | 0.983 | **1.70** | 1.05 | **0.650** | **1.90** | **0.80** | 1.10 |
| **Colon** | Dice | 0.899 | 0.863 | 0.970 | 22.13 | 10.88 | 11.25 | 33.75 | 13.75 | 20.00 |
| | clDice | 0.907 | 0.871 | 0.974 | 23.63 | 9.38 | 14.25 | 37.75 | 11.75 | 26.00 |
| | Hu et al. | 0.902 | 0.876 | 0.972 | 17.25 | 7.75 | 9.50 | 22.00 | 7.00 | 15.00 |
| | Ours | 0.907 | 0.871 | 0.975 | **16.00** | **7.13** | 8.88 | 21.50 | **6.25** | 15.25 |
| **Buildings** | Dice | 0.623 | 0.672 | 0.934 | 286.22 | 275.50 | 10.73 | 162.95 | 151.70 | 11.25 |
| | clDice | 0.632 | **0.693** | 0.931 | 285.60 | 267.98 | 17.63 | 175.50 | 155.05 | 20.45 |
| | Hu et al. | 0.625 | 0.677 | 0.934 | 278.30 | 268.75 | 9.55 | 181.10 | 169.60 | 11.50 |
| | Ours | 0.625 | 0.685 | **0.937** | **244.58** | 235.63 | 8.95 | **118.45** | **107.75** | 10.70 |

**Training of the segmentation networks** For implementation details, e.g., the training splits, please refer to App. I and J. We train all our models for a fixed, dataset-specific number of epochs and evaluate the final model on an unseen test set. We train all models on an Nvidia P8000 GPU using Adam optimizer. We run experiments on a range of alpha-parameters for clDice (Shit et al. (2021)), the Wasserstein matching (Hu et al. (2019)), and TopoMatch; we choose to present the top performing model in Table 1; extended results are given in tables 3, 4, 5,2, 6 in App. H.

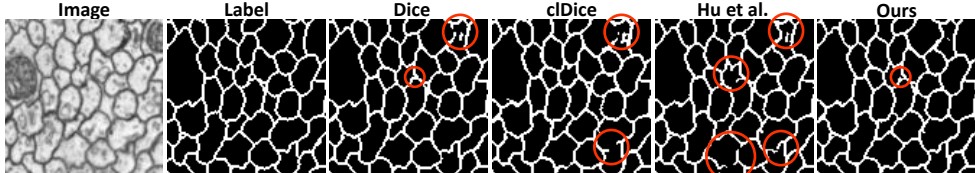

Figure 9: Qualitative Results on CREMI dataset (same models used as in Table 1). Topological errors are indicated by red circles. Our method leads to less topological errors in the segmentation.

## 4.1 RESULTS

**Main Results** Our proposed TopoMatch loss improves the topological accuracy of the segmentations across all datasets (Table 1), irrespective of the choice of hyper-parameters (Table 3) compared to all baselines. We show superior scores for the topological metrics TopoMatch error (T.M.) and Betti number error (Betti) in both dimension-0 and dimension-1. Further, the volumetric metrics (Accuracy, Dice, and clDice) of the segmentations show equivalent, if not superior quantitative results for our method. Our method can be trained from scratch or used to refine pre-trained networks. Importantly, our method improves the topological correctness of curvilinear segmentation problems (Roads, CREMI), blob-segmentation problems (Buildings, Colon), and mixed problems (SynMnist, Elegans). We confidently attribute this to the theoretical guarantees of induced matchings, which hold for the foreground and the background classes in dim-0 and dim-1. For illustration, please consider the Roads and Buildings dataset; essentially, the topology of the background of the Buildings dataset is very similar to the foreground in Roads. I.e., the foreground of the Roads and the background of the Buildings dataset are interesting in dim-1, whereas the background of the roads and the foreground of the Buildings are interesting in dim-0. As our method can efficiently leverage

the topology features of both foreground and background when we apply *sub- and superlevelset-*matching and it is intuitive that our method prevails in both. It is of note that for some datasets, the method by Hu et al. (2019) is the best performing baseline and for some Shit et al. (2021).

Table 2: bothlevel versus superlevel matching of our method on the Elegans dataset and the CREMI dataset. The bothlevel matching appears to have a more pronounced contribution in the scenario of topologically complex background

| level | $\alpha$ | Dice | clDice | Acc. | T.M. | T.M.-0 | T.M.-1 | Betti | Betti-0 | Betti-1 | ARI | VOI |
|---|---|---|---|---|---|---|---|---|---|---|---|---|
| Elegans bothlevel | 0.005 | 0.92 | 0.96 | 0.98 | 1.70 | 1.05 | 0.65 | 1.90 | 0.80 | 1.10 | 0.93 | 0.36 |
| Elegans superlevel | 0.005 | 0.92 | 0.95 | 0.98 | 2.15 | 1.35 | 0.80 | 2.40 | 1.30 | 1.10 | 0.91 | 0.43 |
| CREMI bothlevel | 0.5 | 0.89 | 0.95 | 0.95 | 60.48 | 12.92 | 47.56 | 52.08 | 25.28 | 26.80 | 0.93 | 0.36 |
| CREMI superlevel | 0.5 | 0.89 | 0.95 | 0.96 | 59.20 | 14.40 | 44.80 | 52.24 | 28.16 | 24.08 | 0.93 | 0.35 |

**Ablation experiments** In order to study the effectiveness of the TopoMatch loss, we conduct various ablation experiments. First, we study the effect of the $\alpha$ parameter in our method, see Table 3. We find that increasing $\alpha$ improves the topological metrics. For some datasets, e.g., synMnist, the Dice metric is compromised if $\alpha$ is chosen too big. Therefore, we conclude that $\alpha$ is a tunable and dataset-specific parameter. Ostensibly, the effect of the $\alpha$ parameter cannot be compared directly. Nonetheless, it appears that our method is more robust towards variation in $\alpha$. Second, we study the effect of considering both the foreground and the background (bothlevel) versus solely the foreground (superlevel). We find that bothlevel is particularly useful if the background has a complex topology (e.g. Elegans), whereas superlevel shows a similar performance if the foreground has a more complex topology (e.g. CREMI), see Table 2. Third, we test the effect of pre-training and training from scratch for TopoMatch and the method by Hu et al. (2019). Table 6 shows that our method can be trained from scratch efficiently if not superiorly, whereas the baseline method struggles in that setting – especially on more complex datasets such as CREMI. We attribute this to the spatially correct matching of TopoMatch and its consequences on the gradient (see Sec. 3.2). Training-from-scratch means that there is a lot potential for *false positives* and *false negatives* in the Wasserstein matching (see App. F) since there are a lot noisy features when the network is still uncertain. For example for CREMI we found that the Wasserstein matching matches cycles incorrectly in more then 99 % of the cases, see appendix F.1. Moreover, we observe that TopoMatch optimizes the Wasserstein loss more efficiently (see App. F.2). We also experiment with adding a boundary to images in order to close loops that cross the image border, similar to Hu et al. (2019), and term this **relative TopoMatch**. Table 4 shows a negligible effect on all metrics. For additional ablation and more metrics on the ablation studies, please refer to the App. H. The computational complexity of our matching is $\mathcal{O}(n^3)$, see App. D for details.

## 5 DISCUSSION

**Concluding remarks** In this paper, we propose a rigorous method called *TopoMatch*, which enables the faithful quantification of corresponding topological properties in image segmentation. Herein, our method is the first to guarantee the correct matching of persistence barcodes in image segmentation according to their spatial correspondence. We show that *TopoMatch* is efficient as an interpretable segmentation metric, which can be understood as a sharpened variant of the Betti error. Further, we show how our method can be used to train segmentation networks. Training networks using *TopoMatch* is stable and leads to improvements on all 6 datasets. We foresee vast application potential in challenging tasks such as road network, vascular network and Neuron instance segmentation. We are thus hopeful that our method's theory and experimentation will stimulate future research in this area.

**Limitations** In the general setting of persistent homology of functions on arbitrary topological spaces, there are instances where maps of persistence modules cannot be written as matchings. This is somewhat analogous to the fact that in linear algebra, certain linear transformations cannot be diagonalized. We did not observe any such case in our specific segmentation setting. A theoretical investigation of this question will be the subject of future work. Further, we understand application-specific experimental limitations. Our method's computational complexity is beyond widely used loss functions such as BCE (see App. D); moreover, our current implementation is only available in 2D, whereas the theoretical guarantees trivially generalize to 3D.

## 6 REPRODUCIBILITY STATEMENT

To facilitate understanding of the theory section please refer to the basic definitions in the appendix. The core algorithm (TopoMatch) is available as a python script in the supplementary material and is printed in pseudocode in the appendix, section C. The training details are also described in the appendix, section I. Furthermore, all of our code and all of our experimentation, including baselines and hyperparameters, is available in a public anonymous Github repository [1].

## 7 ETHICS STATEMENT

We, the authors, declare that we strictly adhere to the ICLR Code of Ethics. Our method and experimentation are carried out on (partly modified) public datasets with no known ethical concern associations. Our studies do not involve human subjects and we do not foresee any conflicts of interest and sponsorship, discrimination/bias/fairness concerns, privacy and security issues, legal compliance, and research integrity issues.

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

# A ADDITIONAL TOPOLOGICAL MATCHING ILLUSTRATION

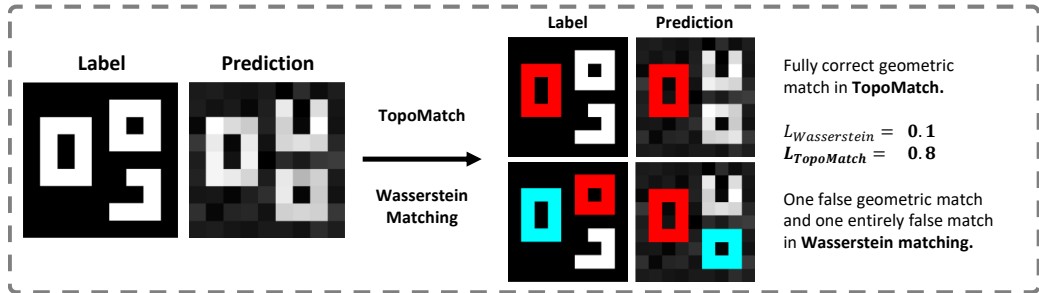

Figure 10: Topological matchings. Illustration of the advantages of our TopoMatch algorithm over the existing *Wasserstein matching* and the *Betti number error* for two exemplary segmentations. On the left side, we depict a prediction-label pair for an image. On the right side, we depict the matched representative cycles in the same color for the Wasserstein matching (bottom row) and *TopoMatch matching* (top row). Our TopoMatch matches the spatially correct features and will penalize the correct features in the loss. Here, the Wasserstein matching mismatches the correctly predicted feature with the erroneously predicted feature, leading to a false loss for the wrongly segmented cycle.

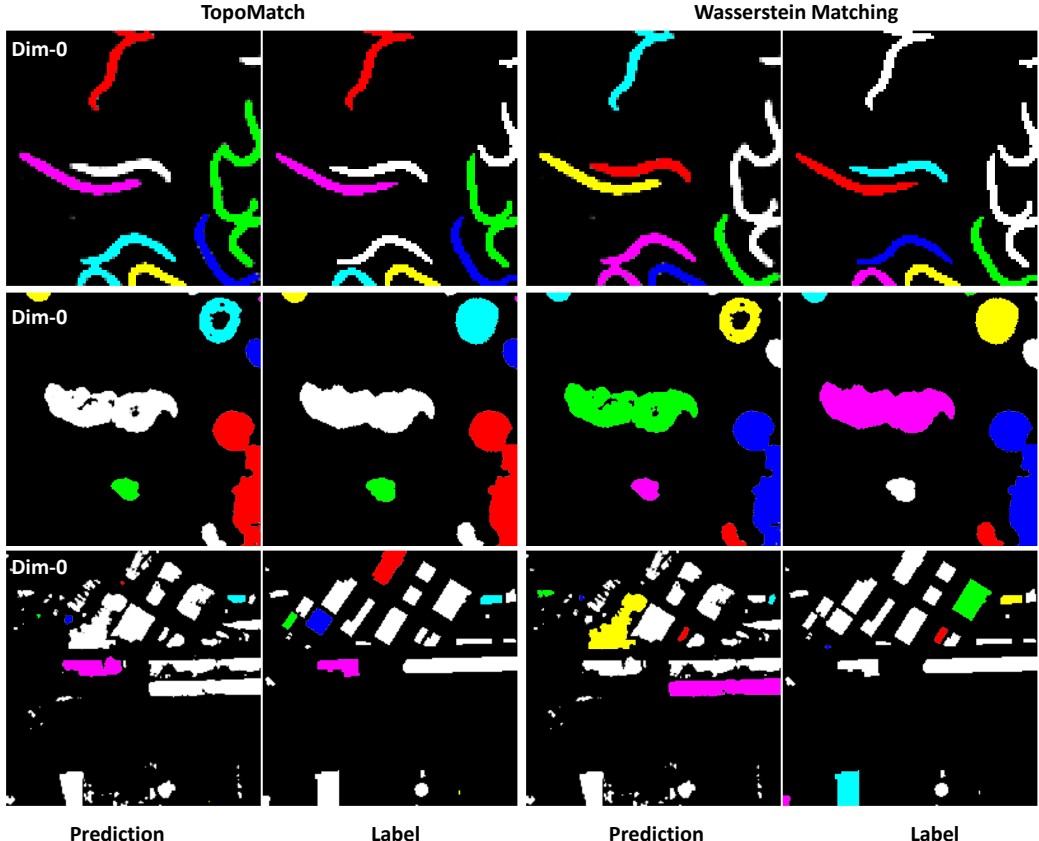

Figure 11: Motivation. Our TopoMatch (induced matching) and the Wasserstein matching (Hu et al. (2019)) for Elegans, Colon and Buildings label-prediction pairs. Here we match the connected components (dim-0). The matched components (according to the matching methods) are represented in the same color. We randomly sample 6 matched cycles in each pair.

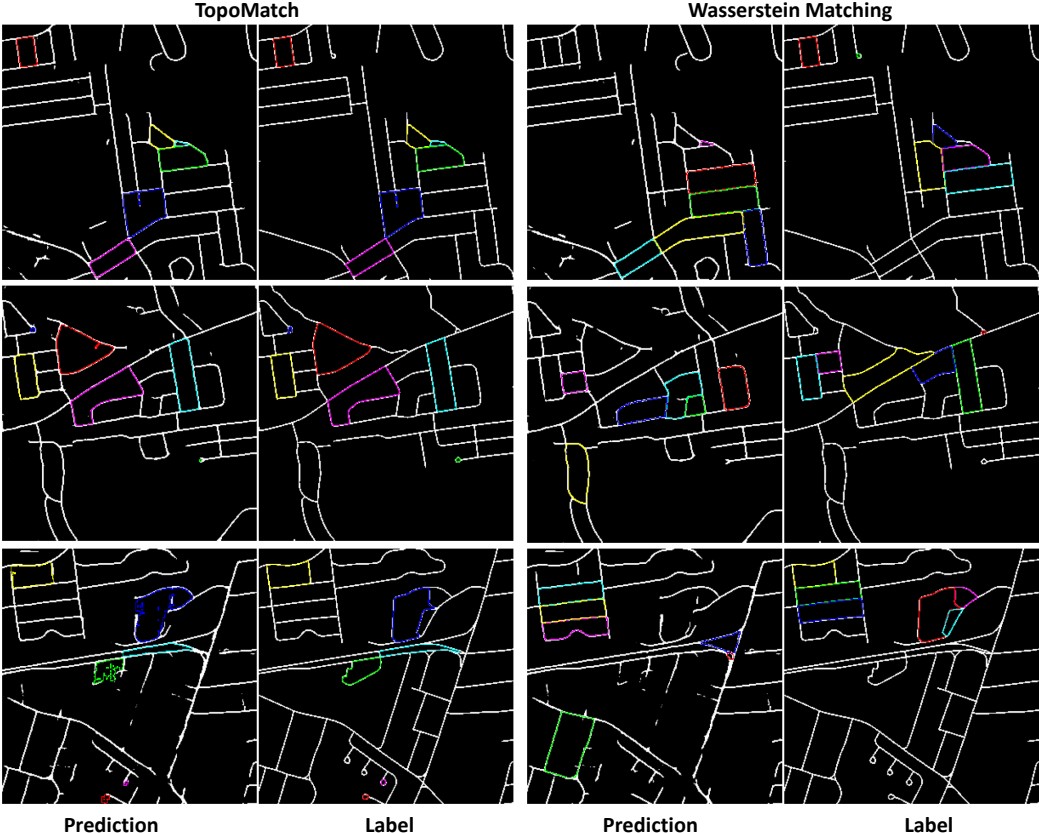

Figure 12: Motivation. Our TopoMatch (induced matching) and the Wasserstein matching (Hu et al. (2019)) for Roads label-prediction pairs. The matched cycles (according to the matching methods) are represented in the same color. We randomly sample 6 matched cycles (dim-1) in each pair. We observe that our method correctly matches the cycles in the first two rows. The third row represents an example early in Training. Here we observe that our method correctly matches some "finished" cycles but also provides a correct matching to the blue and green cycles which still have to be closed. Essentially, one can observe here that our TopoMatch leads to a correct loss.

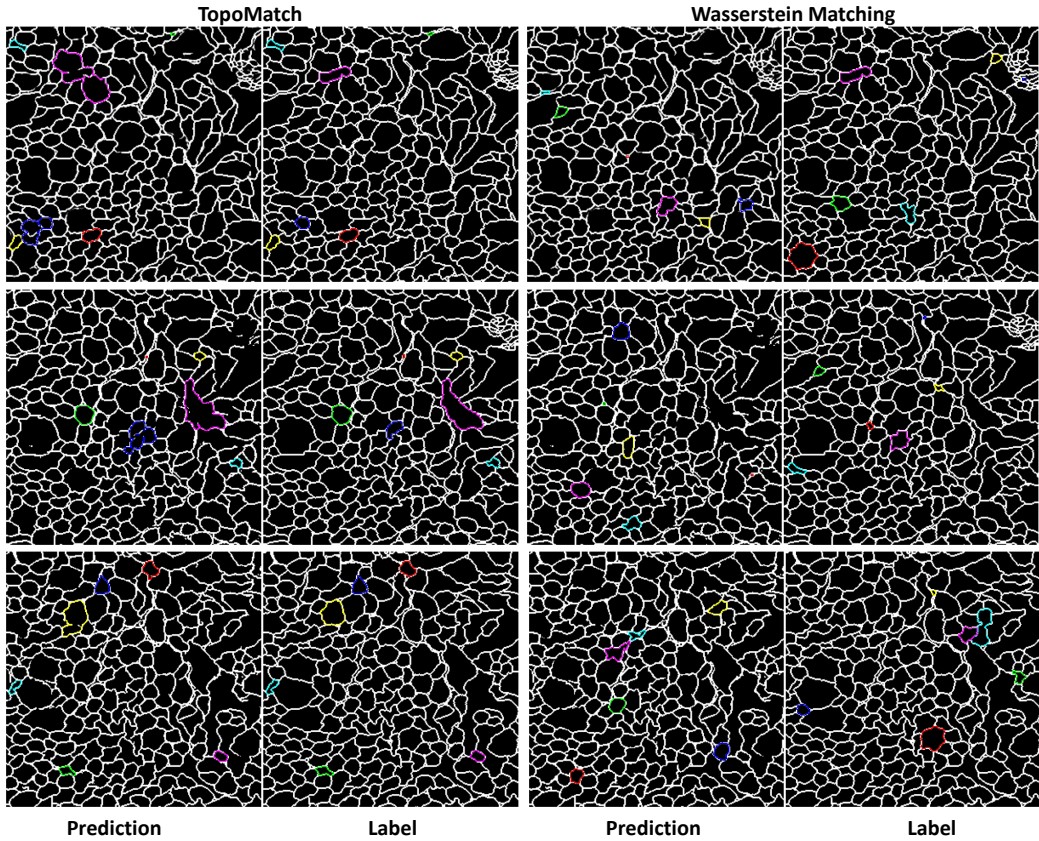

Figure 13: Motivation. Our TopoMatch (induced matching) and the Wasserstein matching (Hu et al. (2019)) for CREMI label-prediction pairs. The matched cycles (according to the matching methods) are represented in the same color. We randomly sample 6 matched cycles in each pair.

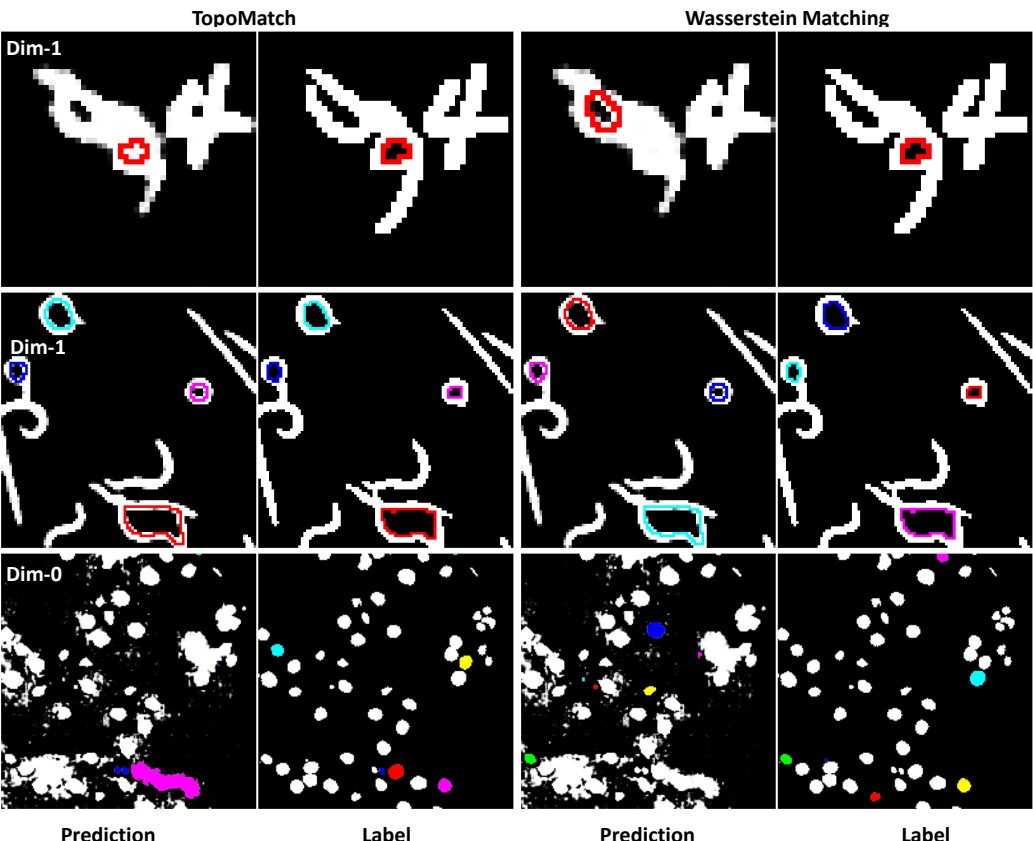

Figure 14: Motivation. Our TopoMatch (induced matching) and the Wasserstein matching (Hu et al. (2019)) for synMnist label-prediction pairs (top row), colon cells (middle row) and the Elegans dataset (lower row). The matched connected components (dim-0) and cycles (dim-1) (according to the matching methods) are represented in the same color. We randomly sample 6 matched cycles in each pair.

# B ADDITIONAL QUALITATIVE RESULTS

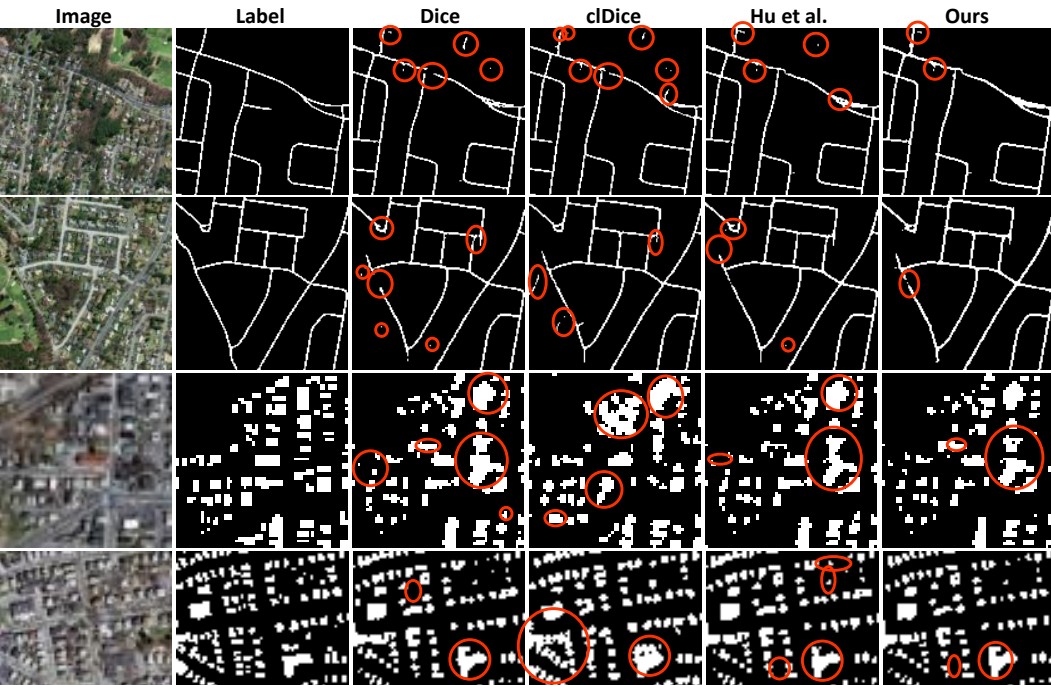

Figure 15: Qualitative Results on Roads and Buildings dataset. Image, Label, and different segmentations (same models as table 1). Topological errors are indicated by red circles. Our method leads to improved topology compared to the baselines.

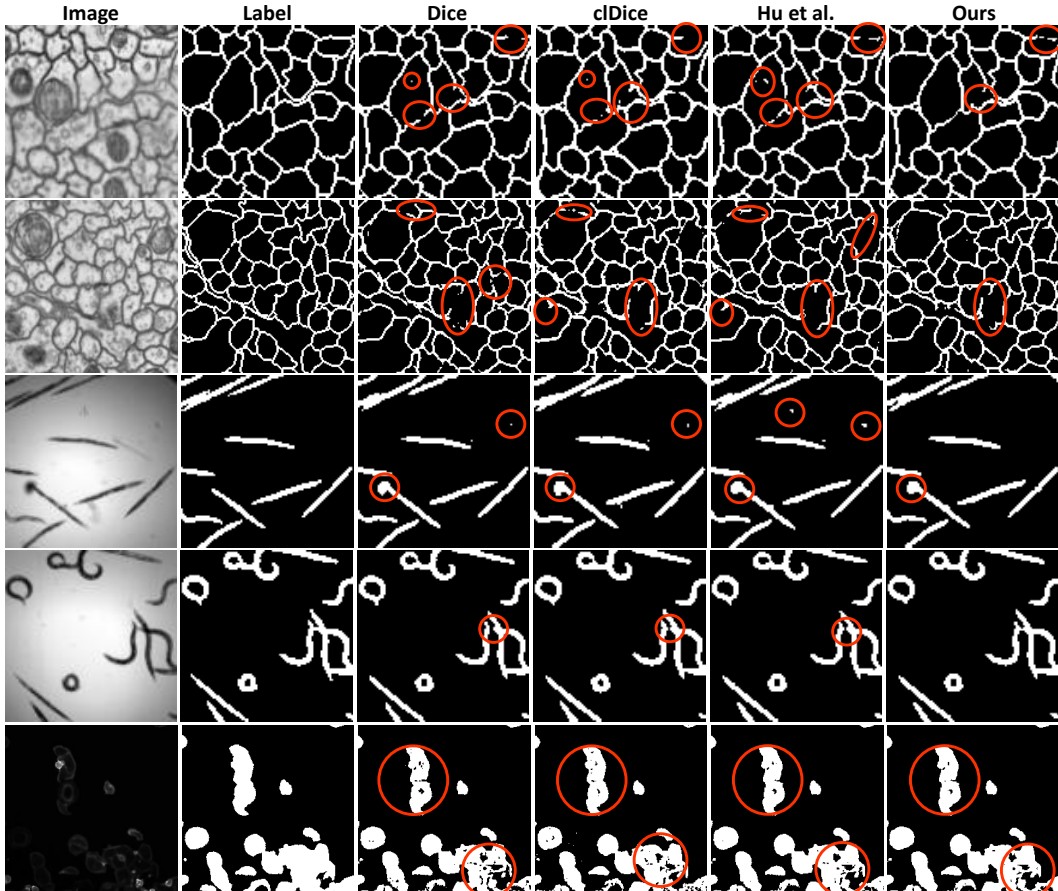

Figure 16: Qualitative Results on CREMI, Elegans and Colon dataset. Image, Label, and different segmentations (same models as table 1). Topological errors are indicated by red circles. Our method leads to improved topology compared to the baselines.

| Image | Label | Dice | clDice | Hu et al. | Ours |
|---|---|---|---|---|---|

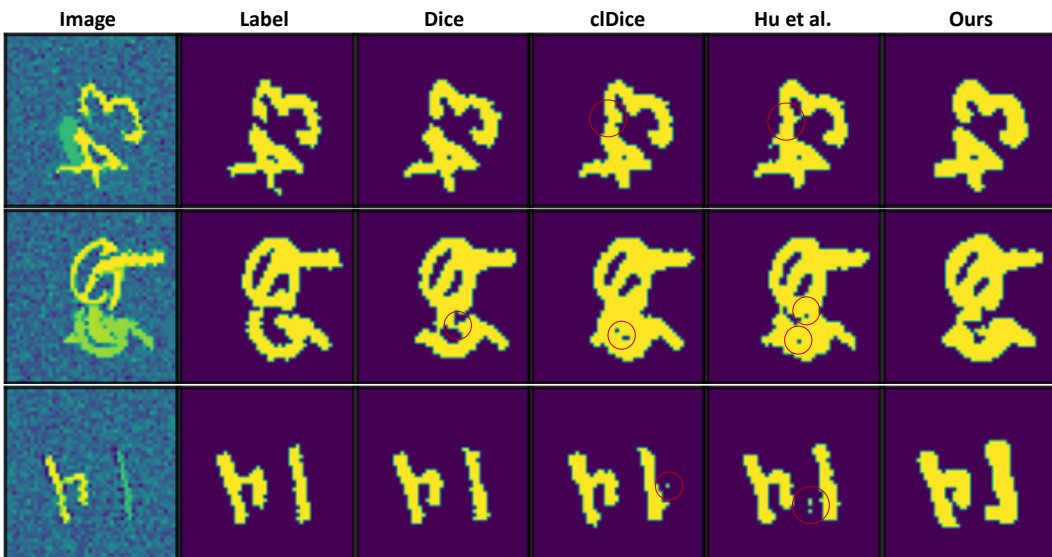

Figure 17: Qualitative Results on SynMnist. Image, Label, and different segmentations (same models as table 1) on examples of the SynMnist testset. Topological errors are indicated by red circles. Our method always segments the correct topology.

## C  DETAIL ALGORITHM

Below, we provide the pseudocode for an efficient realization of the TopoMatch matching. For the computation of the barcodes in dimension-$0$ we leveraged the *Union-Find* datastructure, which is very efficient at managing equivalence classes. Alexander duality allows us to use it in dimension-$1$, as well (see Garin et al. (2020)). Moreover, it can also be used for the computation of the image barcodes in both dimensions. Note that we adapt the Union Find class to manage the birth of equivalence classes. We use *clearing* (as proposed in Bauer (2021)) by keeping track of *critical-edges* and *columns-to-reduce*, in order to reduce the amount of operations during the reductions (see sections 2.2, 2.3).

---

**Algorithm 1:** TopoMatch

---

**Data:** $G, L$
**Option:** $relative = False, filtration = \text{`superlevel'}$
**Result:** $\mathcal{L}_0, \mathcal{L}_1, \mathcal{L}$

1 **begin**
2   **if** *filtration=`superlevel'* **then**       // Construction of comparison image
3      |  $C \leftarrow max(\boldsymbol{G}, \boldsymbol{L})$
4   **else**
5      |  $C \leftarrow min(\boldsymbol{G}, \boldsymbol{L})$
6   **end**
7   $\mathcal{B}(\boldsymbol{G}), \mathbb{D}_G, \boldsymbol{V}_G, \boldsymbol{X}_G \leftarrow CubicalPersistence(\boldsymbol{G}, relative, filtration, True)$;
8   $\mathcal{B}(\boldsymbol{L}), \mathbb{D}_L, \boldsymbol{V}_L, \boldsymbol{X}_L \leftarrow CubicalPersistence(\boldsymbol{L}, relative, filtration, True)$;
9   $\mathcal{B}(\boldsymbol{C}), \mathbb{C}_C, \boldsymbol{V}_C, \boldsymbol{X}_C \leftarrow CubicalPersistence(\boldsymbol{C}, relative, filtration, False)$;
10   $\mathcal{B}(\boldsymbol{G}, \boldsymbol{C}) \leftarrow ImagePersistence(\mathbb{D}_G, \boldsymbol{X}_G, \mathbb{C}_C, \boldsymbol{X}_C)$;
11   $\mathcal{B}(\boldsymbol{L}, \boldsymbol{C}) \leftarrow ImagePersistence(\mathbb{D}_L, \boldsymbol{X}_L, \mathbb{C}_C, \boldsymbol{X}_C)$;
12   $\sigma(\boldsymbol{G}, \boldsymbol{C}) \leftarrow InducedMatching(\mathcal{B}(\boldsymbol{G}, \boldsymbol{C}), \mathcal{B}(\boldsymbol{G}), \mathcal{B}(\boldsymbol{C}))$;
13   $\sigma(\boldsymbol{L}, \boldsymbol{C}) \leftarrow InducedMatching(\mathcal{B}(\boldsymbol{L}, \boldsymbol{C}), \mathcal{B}(\boldsymbol{L}), \mathcal{B}(\boldsymbol{C}))$;
14   $\tau(\boldsymbol{L}, \boldsymbol{G}) = \phi$;       // Initialize matched refined intervals
15   $\mathbb{U}_0, \mathbb{U}_1 = \mathcal{B}(\boldsymbol{G})_0, \mathcal{B}(\boldsymbol{G})_1$ ;    // Initialize unmatched refined intervals for ground truth
16   $\mathbb{V}_0, \mathbb{V}_1 = \mathcal{B}(\boldsymbol{L})_0, \mathcal{B}(\boldsymbol{L})_1$ ;    // Initialize unmatched refined intervals for prediction
17   $\mathcal{L}_0 = \mathcal{L}_1 = 0$ ;       // Initialize TopoMatch loss
18   **for** $d \leftarrow 0$ **to** $1$ **by** $1$ **do**       // Loop over dimension-d
19      **foreach** $m_0 \in \sigma(\boldsymbol{G}, \boldsymbol{C})_d$ **do**
20         **foreach** $m_1 \in \sigma(\boldsymbol{L}, \boldsymbol{C})_d$ **do**
21            **if** $m_0[2] = m_1[2]$ **then**    // Check for same image persistence pair
22               Add $((m_0[0], m_0[2], m_1[0]))$ to $\tau(\boldsymbol{L}, \boldsymbol{G})_d$;
23               Remove $(m_0[0])$ from $\mathbb{U}_d$;
24               Remove $(m_1[0])$ from $\mathbb{V}_d$;
25               Remove $(m_1)$ from $\sigma(\boldsymbol{L}, \boldsymbol{C})_d$;
26               $p, q = m_0[0], m_1[0]$;
27               $I_0, I_1 = \boldsymbol{V}_G(Index2Coord(p[0])), \boldsymbol{V}_G(Index2Coord(p[1]))$ ;    // Map index to value
28               $J_0, J_1 = \boldsymbol{V}_L(Index2Coord(q[0])), \boldsymbol{V}_L(Index2Coord(q[1]))$ ;    // Map index to value
29               $\mathcal{L}_d = \mathcal{L}_d + (I_0 - J_0)^2 + (I_1 - J_1)^2$ ;       // Loss for matched intervals
30               **break**
31            **end**
32         **end**
33      **end**
34      **foreach** $p \in \mathbb{U}_d$ **do**
35         $I_0, I_1 = \boldsymbol{V}_G(Index2Coord(p[0])), \boldsymbol{V}_G(Index2Coord(p[1]))$ ;    // Map index to value
36         $\mathcal{L}_d = \mathcal{L}_d + \frac{(I_0 - I_1)^2}{2}$ ; // Loss for unmatched intervals in ground truth
37      **end**
38      **foreach** $p \in \mathbb{V}_d$ **do**
39         $I_0, I_1 = \boldsymbol{V}_L(Index2Coord(p[0])), \boldsymbol{V}_L(Index2Coord(p[1]))$ ;    // Map index to value
40         $\mathcal{L}_d = \mathcal{L}_d + \frac{(I_0 - I_1)^2}{2}$ ;       // Loss for unmatched intervals in prediction
41      **end**
42   **end**
43   $\mathcal{L} \leftarrow \mathcal{L}_0 + \mathcal{L}_1$ ;       // Total TopoMatch loss
44 **end**

---

```
45  Procedure CubicalPersistence(I, relative, filtration, critical)
46      if relative=True then
47          I ← AddBoundary(I);                          // Add image boundary
48      end
49      V, X, E ← FilterCubeMap(I, filtration); // Valuemap, Indexmap & edges
            are computed using the CubeMap datastructure as in Wagner
            et al. (2012)
50      B(I)₀, B(I)₁ = φ;                        // Initialize refined barcodes
51      C = φ; // Initialize columns-to-reduce for the clearning trick
52      if critical=True then
53          D = φ;   // Initialize critical-edges for the clearing trick
54      end
55      U = UnionFind(#cubes + 1);        // Instantiate a Union-Find class
56      foreach e ∈ E do        // Compute refined intervals in dimension-1
57          b₀, b₁ ← DualBoundary(X, e);    // Find dual boundary of an edge
58          x, y ← U.find(b₀), U.find(b₁);
59          if x = y then
60              Add e to C, continue
61          end
62          b = min(U.getbirth(x), U.getbirth(y));              // Retrieve birth
63          if critical=True then
64              Add e to D;
65          end
66          if (e, b) is valid then                  // Check for positive interval
67              Add (e, b) to B(I)₁
68          end
69          U.union(x, y)
70      end
71      U = UnionFind(#cubes);           // Instantiate a Union-Find class
72      foreach e ∈ C do        // Compute refined intervals in dimension-0
73          b₀, b₁ ← Boundary(X, e);                 // Find boundary of an edge
74          x, y ← U.find(b₀), U.find(b₁);
75          if x = y then
76              continue
77          end
78          b = max(U.getbirth(x), U.getbirth(y));               // Retrieve birth
79          if (e, b) is valid then                  // Check for positive interval
80              Add (e, b) to B(I)₀;
81          end
82          U.union(x, y)
83      end
84      if critical=True then
85          return (B(I)₀, B(I)₁), D, V, X ;          // Return refined barcodes,
                critical-edges, Valuemap & Indexmap
86      else
87          return (B(I)₀, B(I)₁), C, V, X ;          // Return refined barcodes,
                columns-to-reduce, Valuemap & Indexmap
88      end
```

```
89  Procedure ImagePersistence(𝔻, 𝑿_I, ℂ, 𝑿_J)
90  │  ℬ(𝑰,𝑱)_0, ℬ(𝑰,𝑱)_1 = φ ;          // Initialize image persistence pairs
91  │  𝒰 = UnionFind(#cubes) ;           // Instantiate a Union-Find class
92  │  foreach e ∈ ℂ do                  // Compute pairs in dimension-0
93  │  │  b_0, b_1 ← Boundary(𝑿_I, e);          // Find boundary of an edge
94  │  │  x, y ← 𝒰.find(b_0), 𝒰.find(b_1);
95  │  │  if x = y then
96  │  │  │  continue
97  │  │  end
98  │  │  b = max(𝒰.getbirth(x), 𝒰.getbirth(y));         // Retrieve birth
99  │  │  Add (e, b) to ℬ(𝑰,𝑱)_0 ; // All pairs for extended induced matching
    │  │    (see Sec.  2.3)
100 │  │  𝒰.union(x, y)
101 │  end
102 │  𝒰 = UnionFind(#cubes + 1) ;       // Instantiate a Union-Find class
103 │  foreach e ∈ 𝔻 do                  // Compute pairs in dimension-1
104 │  │  b_0, b_1 ← DualBoundary(𝑿_J, e);   // Find dual boundary of an edge
105 │  │  x, y ← 𝒰.find(b_0), 𝒰.find(b_1);
106 │  │  if x = y then
107 │  │  │  continue
108 │  │  end
109 │  │  b = min(𝒰.getbirth(x), 𝒰.getbirth(y));         // Retrieve birth
110 │  │  Add (e, b) to ℬ(𝑰,𝑱)_1 ;       // All intervals for extended induced
    │  │    matching (see Sec.  2.3)
111 │  │  𝒰.union(x, y)
112 │  end
113 │  return (ℬ(𝑰,𝑱)_0, ℬ(𝑰,𝑱)_1) ;      // Return image persistence pairs
114 Procedure InducedMatching(ℬ(𝑰,𝑱), ℬ(𝑰), ℬ(𝑱))
115 │  σ(𝑰,𝑱)_0, σ(𝑰,𝑱)_1 = φ ;     // Initialize matched refined intervals
116 │  for d ← 0 to 1 by 1 do                         // Loop over dimension-d
117 │  │  foreach (a, b) ∈ ℬ(𝑰,𝑱)_d do     // For each image persistence pair
118 │  │  │  m_i, m_j = None;
119 │  │  │  foreach (c, d) ∈ ℬ(𝑰)_d do                 // Match left endpoints
120 │  │  │  │  if c = a then
121 │  │  │  │  │  m_i = (c, d);
122 │  │  │  │  │  break
123 │  │  │  │  end
124 │  │  │  end
125 │  │  │  if m_i = None then           // Skip search if no match found
126 │  │  │  │  continue
127 │  │  │  end
128 │  │  │  foreach (c, d) ∈ ℬ(𝑱)_d do                 // Match right endpoints
129 │  │  │  │  if d = b then
130 │  │  │  │  │  m_j = (c, d);
131 │  │  │  │  │  break
132 │  │  │  │  end
133 │  │  │  end
134 │  │  │  if m_j = None then           // Skip search if no match found
135 │  │  │  │  continue
136 │  │  │  end
137 │  │  │  Add (m_i, (a, b), m_j) to σ(𝑰,𝑱)_d;
138 │  │  │  Remove m_i from ℬ(𝑰)_d;
139 │  │  │  Remove m_j from ℬ(𝑱)_d;
140 │  │  end
141 │  end
142 │  return (σ(𝑰,𝑱)_0, σ(𝑰,𝑱)_1)
```

# D COMPUTATIONAL COMPLEXITY

For a grayscale image represented by a matrix $\boldsymbol{I} \in \mathbb{R}^{M,N}$, we have $n = MN$ number of pixels and form a cubical grid complex of dimension $d = 2$. The computation of the filtration and the boundary matrix can be done efficiently using the CubeMap data structure (see Wagner et al. (2012)) with $\mathcal{O}(3^d n + d^2 n)$ time and $\mathcal{O}(d^2 n)$ space complexity. Computing the barcodes by means of the reduction algorithm requires cubic complexity in the number of pixels $\mathcal{O}(n^3)$ (see Otter et al. (2017)). Despite our empirical acceleration due to the Union-Find class and clearing tricks (as described in Bauer & Schmahl (2022); Bauer (2021)), the order complexity remains $\mathcal{O}(n^3)$. We need $\mathcal{O}(n^2)$ time complexity for computing the final matching and loss. It is noteworthy that Hu et al. (2019) also needs $\mathcal{O}(n^3)$ time complexity to compute the barcode and $\mathcal{O}(n^2)$ for the matching, whereas Shit et al. (2021) requires relatively lower complexity $\mathcal{O}(n)$ due to the overlap based loss formulation.

# E CONVERGENCE OF TOPOMATCH LOSS

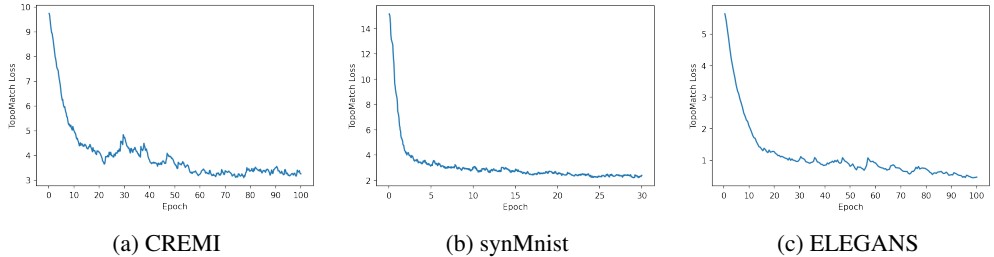

(a) CREMI          (b) synMnist          (c) ELEGANS

Figure 18: Plot of the empirical convergence curves of our TopoMatch loss for the CREMI, MNIST, and ELEGANS datasets. We plot the TopoMatch contribution in the training loss for a varying number of epochs, which is dependent on the dataset size. We show that TopoMatch loss efficiently converges for the different datasets. The absolute magnitude of the loss varies from dataset to dataset because TopoMatch is a real interpretable measure of dim-0 and dim-1 topological features in the training images. E.g. CREMI has a substantially higher number of features, especially cycles, than Elegans, therefore, the absolute magnitude of the loss is likely higher.

# F    WASSERSTEIN MATCHING

The $p$th **Wasserstein distance** is frequently used to measure the difference between persistence diagrams; it is given by

$$d_p(\mathcal{B}_1, \mathcal{B}_2) = \inf_{\gamma} \left( \sum_{q \in \mathrm{Dgm}(\mathcal{B}_1)} \|q - \gamma(q)\|_\infty^p \right)^{1/p}$$

for $p \geq 1$, where $\gamma$ runs over all bijections $\mathrm{Dgm}(\mathcal{B}_1) \to \mathrm{Dgm}(\mathcal{B}_2)$ that respect the dimension. For a likelihood map $\boldsymbol{L} \in [0,1]^{m \times n}$ and a ground truth $\boldsymbol{G} \in \{0,1\}^{m \times n}$, the authors of Hu et al. (2019) adopt this metric to define the **Wasserstein loss**

$$L_{\mathrm{W}}(\boldsymbol{L}, \boldsymbol{G}) = \min_{\gamma} \sum_{q \in \mathrm{Dgm}(\boldsymbol{L})} \|q - \gamma(q)\|_2^2,$$

where $\gamma$ runs over all bijections $\mathrm{Dgm}(\boldsymbol{L}) \to \mathrm{Dgm}(\boldsymbol{G})$ that respect the dimension. The bijection $\gamma_*$ achieving the minimum corresponds to the **Wasserstein matching** $\mathrm{Dgm}(\boldsymbol{L}) \to \mathrm{Dgm}(\boldsymbol{G})$, which minimizes the total distance of matched points. For the represented topological features this means that the matching is purely based on their local contrast within their respective images. Furthermore, note that $\mathrm{Dgm}(\boldsymbol{G})$ contains exclusively the point $(0,1)$ since the entries of $\boldsymbol{G}$ are contained in $\{0,1\}$. Hence, $\gamma_*$ matches points in $\mathrm{Dgm}(\boldsymbol{L})$ representing features in $\boldsymbol{L}$ with enough local contrast in descending order until $\mathrm{Dgm}(\boldsymbol{G})$ runs out of points. This procedure results in a matching of topological features, which potentially exhibit no spatial relation within their respective images (see Fig. 1,8c,19b and App. A) and can have a negative impact on the training of segmentation networks. To see this, we distinguish two cases for a fixed point $q = (q_1, q_2) \in \mathrm{Dgm}(\boldsymbol{L})$:

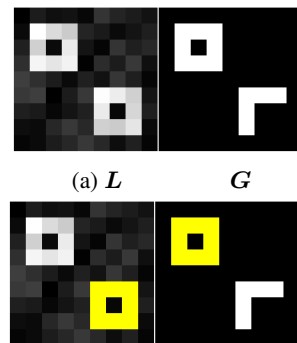

(a) $\boldsymbol{L}$        $\boldsymbol{G}$

(b) Wasserstein matching

Figure 19: (a) A predicted likelihood map $\boldsymbol{L}$ and ground truth segmentation $\boldsymbol{G}$. (b) visualizes the Wasserstein matching $\gamma_*$ (only the yellow cycles are matched), i.e. the top-left cycle in $\boldsymbol{L}$ is a false negative and the bottom-right cycle in $\boldsymbol{L}$ is a false positive.

**case 1: (false positive)** $q$ is matched but there is no spatially corresponding feature in $\boldsymbol{G}$ :

Since $q$ is matched to the point $(0,1) \in \mathrm{Dgm}(\boldsymbol{G})$, the loss $L_{\mathrm{W}}$ will be reduced by decreasing the value $q_1$ and increasing the value $q_2$. Hence, the segmentation network will learn to increase the local contrast of the feature described by the $q$ (see Sec. 3.2), but it should be decreased.

**case 2: (false negative)** $q$ is unmatched but there is a spatially corresponding feature in $\boldsymbol{G}$:

Since $q$ is unmatched, the bijection $\gamma_*$ maps it to its closest point $((q_1 + q_2)/2, (q_1 + q_2)/2)$ on the diagonal $\Delta$ and the loss $L_{\mathrm{W}}$ will be reduced by increasing the value $q_1$ and decreasing the value $q_2$. Hence, the segmentation network will learn to decrease the local contrast of the feature described by $q$ (see Sec. 3.2), but it should be increased.

## F.1    FREQUENCY OF INCORRECT WASSERSTEIN MATCHING

Next, we study how frequently these two cases occur. Assuming that the TopoMatch matching is correct, we evaluate the quality of the Wasserstein matching on the CREMI dataset. Therefor, we choose a segmentation model to obtain label-prediction pairs for every image in the CREMI dataset and compute both matchings. Among the 37243 matched intervals in the barcodes of the predictions by the Wasserstein matching, only 224 have been matched correctly, i.e. it achieves a *precision* of 0.6%.

## F.2    WASSERSTEIN LOSS AS BETTI NUMBER ERROR

For a binarized output $\boldsymbol{P}$ and ground truth $\boldsymbol{G}$, the Wasserstein loss and the Betti number error are closely related. A similar argumentation as in Sec. 3.3 for the TopoMatch loss shows that

$$\beta_{\mathrm{err}}(\boldsymbol{P}, \boldsymbol{G}) = 2 L_{\mathrm{W}}(\boldsymbol{P}, \boldsymbol{G}).$$

A lower Betti number error of a model trained with our TopoMatch loss compared to a model trained with the Wasserstein loss asserts that the TopoMatch loss produces more *faithful* gradients during the training of segmentation networks. Note that, empirically, models trained with TopoMatch loss consistently outperform models trained with Wasserstein los with regard to the Betti number error (see Tables 1,3).

# G PERSISTENCE DIAGRAM AND BARCODES

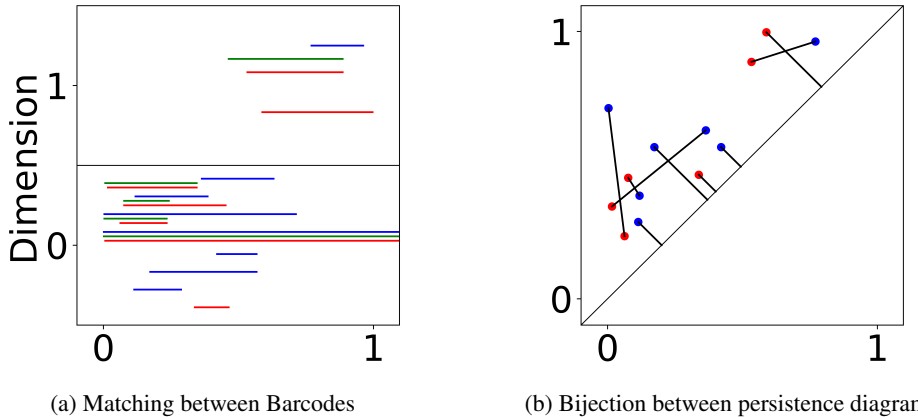

(a) Matching between Barcodes       (b) Bijection between persistence diagram

Figure 20: Illustrations of how to translate a matching between barcodes (a) into a bijection between persistence diagram (b) and vice versa. A red or blue line in (a) is a dot of the same color in (b). In (a), a green interval in between a blue and a red line indicates they are matched. In (b), a line connecting two points indicates that they are matched. For detail, please refer to Section 3.2.

# H ADDITIONAL ABLATION EXPERIMENTS

Table 3: $\alpha$ ablation on the synMnist dataset and the Roads dataset

| | | $\alpha$ | Dice | clDice | Acc. | T.M. | T.M.-0 | T.M.-1 | Betti | Betti-0 | Betti-1 | ARI | VOI |
|---|---|---|---|---|---|---|---|---|---|---|---|---|---|
| Roads | Ours | 0.0005 | 0.670 | 0.706 | 0.974 | 53.958 | 39.896 | 14.063 | 103.917 | 79.375 | 24.542 | 0.643 | 0.847 |
| | | 0.005 | 0.670 | 0.708 | 0.974 | 51.250 | 37.271 | 13.979 | 97.583 | 74.042 | 23.542 | 0.647 | 0.839 |
| | | 0.05 | 0.667 | 0.709 | 0.974 | 45.396 | 31.667 | 13.729 | 85.042 | 62.833 | 22.208 | 0.655 | 0.828 |
| | | 0.5 | 0.663 | 0.713 | 0.972 | 41.500 | 28.146 | 13.354 | 75.083 | 55.792 | 19.292 | 0.690 | 0.791 |
| | clDice | 0.05 | 0.664 | 0.701 | 0.975 | 77.292 | 61.688 | 15.604 | 151.250 | 123.125 | 28.125 | 0.588 | 0.895 |
| | | 0.1 | 0.663 | 0.697 | 0.975 | 81.188 | 65.979 | 15.208 | 158.375 | 131.708 | 26.667 | 0.599 | 0.885 |
| | | 0.25 | 0.667 | 0.701 | 0.975 | 79.188 | 64.125 | 15.063 | 154.208 | 127.917 | 26.292 | 0.622 | 0.879 |
| | | 0.75 | 0.668 | 0.704 | 0.975 | 65.500 | 51.042 | 14.458 | 125.833 | 101.667 | 24.167 | 0.615 | 0.873 |
| | | 0.5 | 0.663 | 0.696 | 0.975 | 78.625 | 63.750 | 14.875 | 152.417 | 127.000 | 25.417 | 0.618 | 0.874 |
| | Hu et al. | 0.0005 | 0.669 | 0.706 | 0.974 | 53.750 | 39.521 | 14.229 | 102.500 | 78.542 | 23.958 | 0.651 | 0.838 |
| | | 0.005 | 0.674 | 0.712 | 0.974 | 50.500 | 36.521 | 13.979 | 95.833 | 72.542 | 23.292 | 0.660 | 0.836 |
| | | 0.05 | 0.669 | 0.707 | 0.974 | 52.896 | 39.208 | 13.688 | 99.875 | 77.917 | 21.958 | 0.654 | 0.832 |
| | | 0.5 | 0.656 | 0.699 | 0.970 | 58.583 | 44.604 | 13.979 | 105.417 | 88.708 | 16.708 | 0.709 | 0.787 |
| synMnist | Ours | 0.0005 | 0.866 | 0.907 | 0.962 | 1.687 | 0.843 | 0.844 | 2.302 | 1.370 | 0.932 | 0.844 | 0.537 |
| | | 0.005 | 0.871 | 0.920 | 0.962 | 1.270 | 0.458 | 0.812 | 1.596 | 0.732 | 0.864 | 0.873 | 0.481 |
| | | 0.05 | 0.849 | 0.916 | 0.955 | 1.140 | 0.265 | 0.875 | 1.348 | 0.426 | 0.922 | 0.868 | 0.491 |
| | | 0.5 | 0.796 | 0.888 | 0.939 | 1.150 | 0.291 | 0.859 | 1.428 | 0.466 | 0.962 | 0.805 | 0.612 |
| | clDice | 0.05 | 0.871 | 0.911 | 0.963 | 1.616 | 0.796 | 0.820 | 2.264 | 1.328 | 0.936 | 0.871 | 0.483 |
| | | 0.1 | 0.872 | 0.912 | 0.963 | 1.642 | 0.816 | 0.826 | 2.320 | 1.384 | 0.936 | 0.862 | 0.506 |
| | | 0.25 | 0.874 | 0.917 | 0.963 | 1.342 | 0.519 | 0.823 | 1.764 | 0.826 | 0.938 | 0.877 | 0.461 |
| | | 0.5 | 0.875 | 0.922 | 0.963 | 1.270 | 0.436 | 0.834 | 1.640 | 0.700 | 0.940 | 0.881 | 0.454 |
| | | 0.75 | 0.869 | 0.921 | 0.961 | 1.196 | 0.401 | 0.795 | 1.580 | 0.622 | 0.958 | 0.888 | 0.429 |
| | Hu et al. | 0.0005 | 0.872 | 0.909 | 0.963 | 1.881 | 1.001 | 0.880 | 2.650 | 1.686 | 0.964 | 0.880 | 0.461 |
| | | 0.005 | 0.870 | 0.908 | 0.962 | 1.787 | 0.911 | 0.876 | 2.498 | 1.514 | 0.984 | 0.864 | 0.504 |
| | | 0.05 | 0.867 | 0.916 | 0.960 | 1.425 | 0.502 | 0.923 | 1.802 | 0.764 | 1.038 | 0.893 | 0.425 |
| | | 0.5 | 0.785 | 0.862 | 0.935 | 1.754 | 0.562 | 1.192 | 1.968 | 0.840 | 1.128 | 0.814 | 0.589 |

Table 4: Relative Frame ablation of our method on the Roads dataset

| | $\alpha$ | Dice | clDice | Acc. | T.M. | T.M.-0 | T.M.-1 | Betti | Betti-0 | Betti-1 | ARI | VOI |
|---|---|---|---|---|---|---|---|---|---|---|---|---|
| relative | 0.0005 | 0.670 | 0.706 | 0.974 | 53.958 | 39.896 | 14.063 | 103.917 | 79.375 | 24.542 | 0.643 | 0.847 |
| | 0.005 | 0.670 | 0.708 | 0.974 | 51.250 | 37.271 | 13.979 | 97.583 | 74.042 | 23.542 | 0.647 | 0.839 |
| | 0.05 | 0.667 | 0.709 | 0.974 | 45.396 | 31.667 | 13.729 | 85.042 | 62.833 | 22.208 | 0.655 | 0.828 |
| | 0.5 | 0.663 | 0.713 | 0.972 | 41.500 | 28.146 | 13.354 | 75.083 | 55.792 | 19.292 | 0.690 | 0.791 |
| non-relative | 0.0005 | 0.669 | 0.706 | 0.974 | 54.729 | 40.521 | 14.208 | 104.542 | 80.542 | 24.000 | 0.654 | 0.835 |
| | 0.005 | 0.671 | 0.709 | 0.974 | 50.250 | 36.479 | 13.771 | 95.167 | 72.458 | 22.708 | 0.661 | 0.829 |
| | 0.005 | 0.669 | 0.712 | 0.973 | 46.104 | 32.271 | 13.833 | 84.708 | 64.042 | 20.667 | 0.675 | 0.818 |
| | 0.5 | 0.661 | 0.711 | 0.972 | 41.417 | 28.167 | 13.250 | 75.583 | 55.833 | 19.750 | 0.695 | 0.787 |

Table 5: dimension-1 and dimensions-0,1 matching ablation for the Hu et al. method on the Roads dataset

| | $\alpha$ | Dice | clDice | Acc. | T.M. | T.M.-0 | T.M.-1 | Betti | Betti-0 | Betti-1 | ARI | VOI |
|---|---|---|---|---|---|---|---|---|---|---|---|---|
| dim-1 | 0.0005 | 0.669 | 0.706 | 0.974 | 53.750 | 39.521 | 14.229 | 102.500 | 78.542 | 23.958 | 0.651 | 0.838 |
| | 0.005 | 0.674 | 0.712 | 0.974 | 50.500 | 36.521 | 13.979 | 95.833 | 72.542 | 23.292 | 0.660 | 0.836 |
| | 0.05 | 0.669 | 0.707 | 0.974 | 52.896 | 39.208 | 13.688 | 99.875 | 77.917 | 21.958 | 0.654 | 0.832 |
| | 0.5 | 0.656 | 0.699 | 0.970 | 58.583 | 44.604 | 13.979 | 105.417 | 88.708 | 16.708 | 0.709 | 0.787 |
| dim-0,1 | 0.0005 | 0.672 | 0.709 | 0.974 | 54.292 | 40.104 | 14.188 | 104.083 | 79.708 | 24.375 | 0.649 | 0.839 |
| | 0.005 | 0.673 | 0.710 | 0.974 | 52.750 | 38.625 | 14.125 | 100.667 | 76.750 | 23.917 | 0.656 | 0.836 |
| | 0.05 | 0.668 | 0.708 | 0.974 | 47.000 | 33.146 | 13.854 | 88.083 | 65.792 | 22.292 | 0.649 | 0.832 |
| | 0.5 | 0.662 | 0.711 | 0.972 | 44.708 | 31.292 | 13.417 | 80.583 | 62.083 | 18.500 | 0.692 | 0.798 |

Table 6: Pretraining vs training from scratch of ours and the Hu et al. method on the Elegans dataset

|  | Training | $\alpha$ | Dice | clDice | Acc. | T.M. | T.M.-0 | T.M.-1 | Betti | Betti-0 | Betti-1 | ARI | VOI |
|---|---|---|---|---|---|---|---|---|---|---|---|---|---|
| CREMI | Ours f. scratch | 0.05 | 0.882 | 0.938 | 0.953 | 65.06 | 13.94 | 51.12 | 45.72 | 27.16 | 18.56 | 0.919 | 0.393 |
|  | Ours pretrained | 0.05 | 0.889 | 0.940 | 0.957 | 65.68 | 14.26 | 51.42 | 64.40 | 27.96 | 36.44 | 0.905 | 0.437 |
|  | Hu et al. f. scratch | 0.05 | 0.880 | 0.932 | 0.953 | 82.76 | 27.92 | 54.84 | 85.60 | 55.28 | 30.32 | 0.905 | 0.436 |
|  | Hu et al. pretrained | 0.05 | 0.895 | 0.942 | 0.960 | 66.40 | 17.76 | 48.64 | 75.36 | 34.96 | 40.40 | 0.909 | 0.425 |
| Elegans | Ours f. scratch | 0.005 | 0.919 | 0.960 | 0.983 | 1.700 | 1.050 | 0.650 | 1.90 | 0.80 | 1.10 | 0.927 | 0.359 |
|  | Ours pretrained | 0.005 | 0.924 | 0.963 | 0.984 | 1.950 | 1.200 | 0.750 | 2.30 | 1.20 | 1.10 | 0.939 | 0.313 |
|  | Hu et al. f. scratch | 0.005 | 0.921 | 0.959 | 0.984 | 2.150 | 1.425 | 0.725 | 2.50 | 1.35 | 1.15 | 0.929 | 0.350 |
|  | Hu et al. pretrained | 0.005 | 0.921 | 0.962 | 0.984 | 2.075 | 1.250 | 0.825 | 2.55 | 1.30 | 1.25 | 0.929 | 0.349 |

## I DATASETS AND TRAINING SPLITS

The full training routine with the complete trainingsets and testsets will be available with our github repository [2]. All our trainings are done on patches of $48 \times 48$ pixels. For the buildings dataset Mnih (2013), we downsample the images to $375 \times 375$ pixels and randomly choose 80 samples for training and 20 for testing. For each epoch, we randomly sample 8 patches from each sample. For the Colon dataset Carpenter et al. (2006); Ljosa et al. (2012), we downsample the images to $256 \times 256$ pixels; we randomly choose 20 samples for training and 4 for testing. For each epoch, we randomly sample 12 patches from each sample. For the CREMI dataset Funke et al. (2019), we downsample the images to $312 \times 312$ pixels; we choose 100 samples for training and 25 for testing. For each epoch, we randomly sample 4 patches from each sample. For the Elegans dataset Ljosa et al. (2012), we crop the images to $96 \times 96$ pixels; we randomly choose 80 samples for training and 20 for testing. For each epoch, we randomly sample 1 patch from each sample. For the synMnist dataset LeCun (1998), we synthetically modify the MNIST dataset to an image size of $48 \times 48$ pixels; please see our GitHub repository for details; we train on 4500 full, randomly chosen images and use 1500 for testing. For the Roads dataset Mnih (2013), we downsample the images to $375 \times 375$ pixels; we randomly choose 100 samples for training and 24 for testing. For each epoch, we randomly sample 8 patches from each sample.

## J NETWORK SPECIFICATIONS

We use the following notation:

1. In($input\ channels$), Out($output\ channels$), BI($output\ channels$) present input, output, and bottleneck information (for U-Net);
2. C($filter\ size, output\ channels$) denote a convolutional layer followed by $ReLU$ and batch-normalization;
3. U($filter\ size, output\ channels$) denote a trans-posed convolutional layer followed by $ReLU$ and batch-normalization;
4. $\downarrow 2$ denotes maxpooling;
5. $\oplus$ indicates concatenation of information from an encoder block.

### J.1 UNET CONFIGURATION-I

We use this configuration for CREMI, synthMNIST, Colon and Elegans dataset. This is a lightweight U-net which has sufficient expressive power for these datasets.

**ConvBlock :** $C_B(3, out\ size) \equiv C(3, out\ size) \rightarrow C(3, out\ size) \rightarrow\downarrow 2$

**UpConvBlock:** $U_B(3, out\ size) \equiv U(3, out\ size) \rightarrow \oplus \rightarrow C(3, out\ size)$

**Encoder :** $IN(1/3\ ch) \rightarrow C_B(3, 16) \rightarrow C_B(3, 32) \rightarrow C_B(3, 64) \rightarrow C_B(3, 128) \rightarrow C_B(3, 256) \rightarrow B(256)$

---

[2] github/anonymous

**Decoder :** $B(256) \rightarrow U_B(3, 256) \rightarrow U_B(3, 128) \rightarrow U_B(3, 64) \rightarrow U_B(3, 32) \rightarrow U_B(3, 16) \rightarrow Out(1)$

## J.2 UNET CONFIGURATION-II

We had to choose a different U-Net architecture for the road and building dataset because we realized that a larger model is needed to learn useful features for this complex task.

**ConvBlock :** $C_B(3, out\ size) \equiv C(3, out\ size) \rightarrow C(3, out\ size) \rightarrow \downarrow 2$

**UpConvBlock:** $U_B(3, out\ size) \equiv U(3, out\ size) \rightarrow \oplus \rightarrow C(3, out\ size)$

**Encoder :** $IN(3\ ch) \rightarrow C_B(3, 64) \rightarrow C_B(3, 128) \rightarrow C_B(3, 256) \rightarrow C_B(3, 512) \rightarrow C_B(3, 1024) \rightarrow B(1024)$

**Decoder :** $B(1024) \rightarrow U_B(3, 1024) \rightarrow U_B(3, 512) \rightarrow U_B(3, 256) \rightarrow U_B(3, 128) \rightarrow U_B(3, 64) \rightarrow Out(1)$

## K EVALUATION METRICS

We evaluate our experiments using a set of topological and pixel-based metrics. The metrics are computed with respect to the binarized predictions. Here, TopoMatch constitutes the most meaningful quantification, see section 3.3. We calculate the TopoMatch metric for dimension-0 (T.M.-0) and dimension-1 (T.M.-1) as well as their sum (T.M.). Furthermore, we implement the **Betti number error** for dimension-0 (Betti 0), dimension-1 (Betti 1), and their sum (Betti):

$$\beta_{\text{err}}(\boldsymbol{P}, \boldsymbol{G}) = \sum_{d=0}^{\infty} |\beta_d(D(\boldsymbol{P})_{0.5}) - \beta_d(D(\boldsymbol{G})_{0.5})|$$

It computes the Betti numbers of both foregrounds and sums up their absolute difference in each dimension, i.e. it compares the topological complexity of the foregrounds. It is important to consider the dimensions separately since they have different relevance on different datasets. E.g., Roads has many 1-cycles, whereas Buildings has many 0-cycles (connected components).

Additionally, we use the traditional Dice metric and Accuracy, which describe the in total correctly classified pixels, as well as the clDice metric from Shit et al. (2021). Here, we calculate the clDice between the volumes and the skeleta, extracted using the *skeletonize* function of the skimage python-library. We compute all metrics on the individual test images of their respective size (without patching) and take the mean across the whole testset.

## L BASIC DEFINITIONS AND TERMINOLOGY

### L.1 CUBICAL COMPLEXES

A $d$-dimensional **(cubical) cell** in $\mathbb{R}^n$ is the Cartesian product $c = \prod_{j=1}^{n} I_j$ of intervals $I_j = [a_j, b_j]$ with $a_j \in \mathbb{Z}$, $b_j \in \{a_j, a_j + 1\}$ and $d \in \{0, \dots, n\}$ is the number of non-degenerate intervals among $\{I_1, \dots, I_d\}$.

If $c$ and $d$ are cells and $c \subseteq d$, we call $c$ a **face** of $d$ of **codimension** $\dim(d) - \dim(c)$. A face of codimension one is also called a **facet**.

A $d$-dimensional **(cubical) complex** in $\mathbb{R}^n$ is a finite set of cubical cells in $\mathbb{R}^n$ with maximal dimension $d$ that is closed under the face relation, i.e., if $d \in K$ and $c$ is a face of $d$, then $c \in K$. Furthermore we call a cubical complex $K' \subseteq K$ a **subcomplex** of $K$.

A **filtration** of a cubical complex $K$ is given by a family $(K_r)_{r \in \mathbb{R}}$ of subcomplexes of $K$, which satisfies:

(1) $K_r \subseteq K_s$ for all $r \leq s$,

(2) $K = K_r$ for some $r \in \mathbb{R}$.

A **filtered (cubical) complex** $K_*$ is a cubical complex $K$ together with a nested sequence of subcomplexes, i.e., a sequence of complexes

$$\emptyset = K_0 \subseteq K_1 \dots \subseteq K_m = K.$$

A function $f \colon K \to \mathbb{R}$ on a cubical complex is said to be **order preserving** if $f(c) \leq f(d)$ for a face $c$ of a cell $d$.

### L.2 HOMOLOGY

A **chain complex** $C_*$ consists of a family $\{C_d\}_{d \in \mathbb{Z}}$ of vector spaces and a family of linear maps $\{\partial_d \colon C_d \to C_{d-1}\}_{d \in \mathbb{Z}}$ that satisfy $\partial_{d-1} \circ \partial_d = 0$.

A map $f \colon K \to K'$ between cubical complexes is said to be **cubical** if it respects the face relation, i.e., $f(c)$ must be a face of $f(d)$ in $K'$ if $c$ is a face of $d$ in $K$.

### L.3 HOMOLOGY OF CUBICAL COMOPLEXES

*Homology* is a powerful concept involving local computations to capture information about the global structure of topological spaces. It assigns a sequence of abelian groups to a space which encode its topological features in all dimensions. A feature in dimension-0 describes a connected component, in dimension-1, it describes a loop, and in dimension-2, it describes a cavity. It also relates these features between spaces by inducing homomorphisms between their respective homology groups. We briefly introduce the homology of cubical complexes with coefficients in $\mathbb{F}_2$. For more details, we refer to Kaczynski et al. (2004).

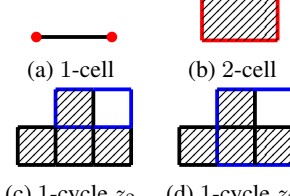

(a) 1-cell     (b) 2-cell

(c) 1-cycle $z_2$     (d) 1-cycle $z_2$

For $d \in \mathbb{Z}$, we denote by $K_d$ the set of $d$-dimensional cells in a cubical complex $K$. The $\mathbb{F}_2$-vector space $C_d(K)$ freely generated by $K_d$ is the **chain group** of $K$ in degree $d$. We can think of the elements in $C_d(K)$ as sets of $d$-dimensional cells and call them **chains**. These chain groups are connected by linear **boundary maps** $\partial_d \colon C_d(K) \to C_{d-1}(K)$, which map a cell to the sum of its faces of codimension 1 and are extended linearly to all of $C_d(K)$. The **cubical chain complex** $C_*(K)$ is given by the pair $(\{C_d(K)\}_{d \in \mathbb{Z}}, \{\partial_d\}_{d \in \mathbb{Z}})$. We denote by $Z_d(K) = \ker \partial_d$ the subspace of **cycles** and by $B_d(K) = \operatorname{im} \partial_{d+1}$ the subspace of **boundaries** in $C_d(K)$. Since $\partial_{d-1} \circ \partial_d = 0$, every boundary is a cycle and the **homology group** of $K$ in degree $d$ is defined by the quotient space $H_d(K) := Z_d(K)/B_d(K)$. In other words, $H_d(K)$ consists of equivalence classes of $d$-cycles and two $d$-cycles

Figure 21: (a) and (b) show cells and their boundary (red). (c) and (d) visualize two homologous 1-cycles (blue) in a cubical complex.

$z_1, z_2$ are equivalent (**homologous**) if their difference is a boundary. For convenience, we define $H_*(K) = \bigoplus_{d \in \mathbb{Z}} H_d(K)$. Note that the homology groups still carry the structure of a $\mathbb{F}_2$-vector space and their dimension $\beta_d(K) = \dim_{\mathbb{F}_2}(H_d(K))$ is the $d$th **Betti number** of $K$.

Homology does not only act on spaces; it also acts on maps between spaces. Therefor, a *cubical* map $f \colon K \to K'$ induces a linear map $C_*(f) \colon C_*(K) \to C_*(K')$, by mapping a cell $c \in K$ with $\dim(f(c)) = \dim(c)$ to $f(c)$ and extending this assignment linearly to all of $C_*(K)$. Then $C_*(f)$ descends to a linear map $H_*(f) \colon H_*(K) \to H_*(K')$ in homology since $\partial_* \circ C_*(f) = C_*(f) \circ \partial_*$.

### L.4   Persistence modules

A **persistence module** $M$ consists of a family $\{M_r\}_{r \in \mathbb{R}}$ of vector spaces, which are connected by linear **transition maps** maps $M_{r,s} \colon M_r \to M_s$ for all $r \leq s$, such that

(1)  $M_{r,r} = \mathrm{id}_{M_r}$ for all $r \in \mathbb{R}$,

(2)  $M_{s,t} \circ M_{r,s} = M_{r,t}$ for $r \leq s \leq t$.

$M$ is said to be **pointwise finite-dimensional** (p.f.d.) if $M_r$ is finite-dimensional for every $r \in \mathbb{R}$.

A basic example of a persistence module is an **interval module** $C(I)$ for a given interval $I \subseteq \mathbb{R}$. It consists of vector spaces

$$C(I)_r = \begin{cases} \mathbb{F}_2 & \text{if } r \in I, \\ 0 & \text{otherwise.} \end{cases}$$

and transition maps

$$C(I)_{r,s} = \begin{cases} \mathrm{id}_{\mathbb{F}_2} & \text{if } r, s \in I, \\ 0 & \text{otherwise.} \end{cases}$$

for $r \leq s$.

A **morphism** $\Phi \colon M \to N$ between persistence modules is a family $\{\Phi_r \colon M_r \to N_r\}_{r \in \mathbb{R}}$ of linear maps, such that for all $r \leq s$ the following diagram commutes:

$$
\begin{array}{ccc}
M_r & \xrightarrow{M_{r,s}} & M_s \\
\Phi_r \downarrow & & \downarrow \Phi_s \\
N_r & \xrightarrow{N_{r,s}} & N_s
\end{array}
$$

We call $\Phi$ an **isomorphism** (resp. **monomorphism**, **epimorphism**) of persistence modules if $\Phi_r$ is a isomorphism (resp. monomorphism, epimorphism) of vector spaces for all $r \in \mathbb{R}$.

For a family $\{M_i\}_{i \in I}$ of persistence modules, the **direct sum** $\bigoplus_{i \in I} M_i$ is the persistence module consisting of vector spaces $(\bigoplus_{i \in I} M_i)_r = \bigoplus_{i \in I}(M_i)_r$ for all $r \in \mathbb{R}$ and transition maps $(\bigoplus_{i \in I} M_i)_{r,s} = \bigoplus_{i \in I}(M_i)_{r,s}$ for all $r \leq s \in \mathbb{R}$.

A **multiset** $X$ consists of a set $|X|$ together with a **multiplicity function** $\mathrm{mult}_X \colon |X| \to \mathbb{N} \cup \{\infty\}$. Equivalently it can be represented by its **underlying set** $\amalg X = \bigcup_{x \in |X|} \coprod_{i=1}^{\mathrm{mult}_X(x)} \{x\}$. We say $X$ is finite if its underlying set $\amalg X$ is finite and its cardinality $\#X$ is given by the cardinality of its underlying set.

Let $K_*$ be a filtered cubical complex and $L_*$ a cell-wise refinement according to the compatible ordering $c_1, \ldots, c_l$ of the cells in $K$. The **boundary matrix** $B \in \mathbb{F}_2^{l \times l}$ of $L_*$ is given entry-wise by

$$\boldsymbol{B}_{i,j} = \begin{cases} 1 & \text{if } \sigma_i \text{ is a facet of } \sigma_j, \\ 0 & \text{otherwise.} \end{cases}$$

### L.5   Matchings

A **map** $f \colon X \to Y$ between multisets is a map $f \colon \amalg X \to \amalg Y$ between their underlying sets.

A **matching** $\sigma \colon X \to Y$ between multisets is a bijection $\sigma \colon X' \to Y'$ for some multisets $X', Y'$ that satisfy $\amalg X' \subseteq \amalg X$ and $\amalg Y' \subseteq \amalg Y$. We call

- $\mathrm{coim}(\sigma) = X'$ the **coimage** of $\sigma$,
- $\mathrm{im}(\sigma) = Y'$ the **image** of $\sigma$,
- $\ker(\sigma) = X \setminus X'$ the **kernel** and of $\sigma$,
- $\mathrm{coker}(\sigma) = Y \setminus Y'$ the **cokernel** of $\sigma$.

For a morphism $\Phi \colon M \to N$ of persistence modules, the **image** of $\Phi$ is the persistence module $\mathrm{im}(\Phi)$, with $\mathrm{im}(\Phi)_r = \mathrm{im}(\Phi_r)$ and transition maps $\mathrm{im}(\Phi)_{r,s} = N_{r,s}|_{\mathrm{im}(\Phi_r)} \colon \mathrm{im}(\Phi_r) \to \mathrm{im}(\Phi_s)$ for $r, s \in \mathbb{R}$.

Let $M, N$ be persistence modules. We call $M$ a **(persistence) submodule** of $N$ if $M_r$ is a subspace of $N_r$ for every $r \in \mathbb{R}$ and the inclusions $i_r \colon M_r \hookrightarrow N_r$ assemble to a persistence map $i = (i_r)_{r \in \mathbb{R}}$. In this case we write $M \subseteq N$.

The **composition** of two matchings $X \xrightarrow{\sigma_1} Y \xrightarrow{\sigma_2} Z$ is given by the composition of the bijections

$$\sigma_1^{-1}(Y') \xrightarrow{\sigma_1} Y' \xrightarrow{\sigma_2} \sigma_2(Y'),$$

with $Y' = \amalg \, \mathrm{im}(\sigma_1) \cap \amalg \, \mathrm{coim}(\sigma_2)$.

A persistence module $M$ is said to be **staggered** if every real number $r \in \mathbb{R}$ occurs at most once as endpoint of an interval in $\mathcal{B}(M)$.

