# OpenReview forum: " Topologically faithful image segmentation via induced matching of persistence barcodes"
_ICLR.cc/2023/Conference — Submitted to ICLR 2023_

### Official Review · Reviewer_GXoX · 2022-10-24

**Confidence:** 5
**Correctness:** 3
**Technical Novelty And Significance:** 3
**Empirical Novelty And Significance:** 3
**Recommendation:** 6

**Clarity, Quality, Novelty And Reproducibility:**

Clarity: This paper is well written and easy to follow.

Quality: This paper overcomes the issues of Hu et al.(2019) and achieves better topological results.

Novelty: This paper naturally extends Hu et al.(2019) and contributes to the community.

Reproducibility: I am optimistic to reproduce the main results of this paper as the paper provides lots of details.

**Strength And Weaknesses:**

**Strengths**:

1. This paper achieves a spatially correct matching between the topological features (persistence barcodes) of label (ground truth) and prediction (output of the neural network).

2. This paper proposes an efficient algorithm to compute TopoMatch for images. This paper shows that TopoMatch is an interpretable metric to evaluate the topological correctness of segmentations. It is better than the Betti number as an evaluation metric.

3. Matching L & G directly has spatial matching issues; using induced matching via map C helps with spatially correct matching.

4. This paper uses both background and/or foreground class topology, depending on dataset.

**Weaknesses**:

1. Values in Table 1 are not correctly bold (worse performing is also bold).

2. The proposed method achieves worse pixel-wise performances compared with baselines.

**Summary Of The Paper:**

This paper shows how induced matchings guarantee the spatially correct matching between barcodes in a segmentation setting. Furthermore, this paper proposes an efficient algorithm to compute TopoMatch for images. Also this paper shows that TopoMatch is an interpretable metric to evaluate the topological correctness of segmentations. Moreover, this paper demonstrates how induced matchings can be used to train segmentation networks and improve the topological correctness of the segmentations across all 6 baseline datasets while preserving volumetric segmentation performance.

**Summary Of The Review:**

This paper is a non-trivial extension of Hu et al.(2019) and achieves better topological performances.

---

> ### Author Response · Authors · 2022-11-11
> **Author response**
>
> Dear Reviewer GXoX,
>
> we sincerely appreciate your positive remarks and attentive comments.
>
> First, we want to apologize for the obvious highlighting mistakes in our tables. We have corrected them, see below, and will update the manuscript.
> Moreover, we would like to elaborate on the comment on worse pixel-wise performances compared with the baselines. We evidently acknowledge that the mean volumetric scores (Dice, clDice, Accuracy) are not always higher in our method. However, none of the methods is consistently the best in volumetric scores.
> To compare beyond the mean score, we now study the standard deviation for all metrics and update the boldfacing accordingly.
>
> Instead of plainly highlighting the best numbers, we now consider the standard deviation of the metrics across the test set. We now boldface the highest mean value in a dataset if it is at least 1/8 of a standard deviation away from the next best-performing method. This method illustrates the magnitude of the differences in metrics.
> We show that the performance differences on volumetric metrics are extremely minor, whereas the performance of our method in topological metrics is consistently superior to all baselines. Note that naturally, the clDice Loss function optimizes the clDice metric.
> Importantly, our experimentation now depicts the superiority of our method. Please note that we only show 3 datasets in this reply because of space constraints. The full table is in the updated manuscript.
>
> We would kindly like to ask if you see a need for any clarifications or additions beyond our comments above that we can provide in the remaining rebuttal period to further improve your impression.
>
>
>
>
> ## Table 1
>
>
> | ****Loss****    | ****Dice**** | ****clDice****   | ****Acc.****     | ****T.M.****      | ****T.M.-0****    | ****T.M.-1****   | ****Betti****     | ****Betti-0****   | ****Betti-1****  |
> |-----------------|--------------|------------------|------------------|-------------------|-------------------|------------------|-------------------|-------------------|------------------|
> |**CREMI**|--------------|------------------|------------------|-------------------|-------------------|------------------|-------------------|-------------------|------------------|
> | **Dice**        | 0.894        | 0.939            | 0.959            | 74.82             | 19.84             | 54.98            | 114.12            | 39.12             | 75.00            |
> | **clDice**      | 0.879        | **0.944**        | 0.952            | 73.52             | 17.18             | 56.34            | 103.92            | 33.64             | 70.28            |
> | **Hu et al.**   | 0.888        | 0.935            | 0.957            | 81.24             | 22.12             | 59.12            | 118.16            | 43.68             | 74.48            |
> | **Ours**        | 0.893        | 0.941            | 0.959            | **64.90**         | **15.50**         | **49.40**        | **79.16**         | **30.36**         | **48.80**        |
> | **Roads** | ----------   | ---------------- | ---------------- | ----------------- | ----------------- | ---------------- | ----------------- | ----------------- | ---------------- |
> | **Dice**        | 0.663        | 0.698            | 0.974            | 58.90             | 43.52             | 15.38            | 113.96            | 86.54             | 27.42            |
> | **clDice**      | 0.668        | 0.704            | 0.975            | 65.50             | 51.04             | 14.46            | 125.83            | 101.67            | 24.17            |
> | **Hu et al.**   | 0.674        | 0.712            | 0.974            | 50.50             | 36.52             | 13.98            | 95.83             | 72.54             | 23.29            |
> | **Ours**        | 0.663        | 0.713            | 0.972            | **41.50**         | **28.15**         | 13.35            | **75.08**         | **55.79**         | **19.29**        |
> | **synMnist** | ----------   | ---------------- | ---------------- | ----------------- | ----------------- | ---------------- | ----------------- | ----------------- | ---------------- |
> | **Dice**        | 0.871        | 0.907            | 0.962            | 1.849             | 0.979             | 0.870            | 2.590             | 1.674             | 0.916            |
> | **clDice**      | 0.875        | 0.921            | 0.963            | 1.270             | 0.436             | 0.834            | 1.640             | 0.700             | 0.940            |
> | **Hu et al.**   | 0.866        | 0.915            | 0.960            | 1.425             | 0.502             | 0.923            | 1.802             | 0.764             | 1.038            |
> | **Ours**        | 0.849        | 0.915            | 0.954            | **1.140**         | **0.265**         | 0.875            | **1.348**         | **0.426**         | 0.922            |

---

> > ### Comment · Reviewer_GXoX · 2022-12-07
> > **Thanks very much for your response**
> >
> > Thanks very much for your answers to my queries. Please try to incorporate the revised table into the final version. The authors have addressed my concerns and I tend to accept this paper.

---

### Official Review · Reviewer_jgui · 2022-10-24

**Confidence:** 2
**Correctness:** 3
**Technical Novelty And Significance:** 2
**Empirical Novelty And Significance:** 2
**Recommendation:** 5

**Clarity, Quality, Novelty And Reproducibility:**

The clarity of this paper could be further improved, as mentioned above. The writing quality has room to improve, and the experimental results do not show impressive improvement or a good balance of segmentation accuracy and topology correctness. Also, the novelty is limited and the reproducibility of this work depends on the release of its source code.

**Strength And Weaknesses:**

Strengths:

 +) This paper works on an interesting and important task, the topological correctness of segmented masks.

 +) The proposed TopoMatch sounds reasonable according to its description and experimental results.

Weakness:

 -) The writing of this paper could be greatly improved for a better understanding. For instance, more detailed explanations of figures are desired. In Fig.1, it says the same color means the matched cycles. How about white color? A further explanation of Fig.1 in the main text is missing, which takes time to get the information this figure conveys.  Also, the main content should be self-contained and understandable without the need of an appendix. However, in Section 3.3, the reader needs to read the appendix to understand #ker and #coker.

-) More explanation of the physical meaning of TopoMatch is desired. What is p in the TopoMatch loss equation?

-) The comparison image is defined as C = min(L, G) in the main text, while in the caption of Fig.7, it is defined as C = max(L, G). Why the TopoMatch matching is the composition of two induced matching?

-) The contribution of this paper should be explicitly pointed out. It seems like the main contribution is the use of two-induced matching to define a new metric, which seems to be incremental.

-) The experimental results show that the correction of the topology of segmentation hinders the improvement of segmentation accuracy. Also, once the dice score is over 0.9, the topology correctness, even the spatial correspondences, would not be a big issue, as demonstrated by the Betti number. Is it necessary to have such a complex term in the loss function? Or can we choose a good segmentation mode and just use the Betti number as a constraint? It seems like improving the segmentation accuracy is another good way to correct the topological issues of segmentation. So, in what situations, we do need the topology correctness term?

-) Perhaps more descriptions on the meat of this paper while concisely introducing some background and making connections to the method proposed in this paper.

**Summary Of The Paper:**

This paper provides a metric TopoMatch to evaluate the topological correctness of image segmentation masks. TopoMatch matching is based on induced matching and its advantage over Wasserstein matching is its consideration of spatial correspondence of topological features. Also, TopoMatch can be used as a loss to improve the segmentation results. This method is evaluated on six datasets. Although the segmentation accuracy is not always the best, its topological correctness performs best in all experiments.

**Summary Of The Review:**

Overall, the current shape of this paper makes it not easy to be understood. The experimental results do not convince me to choose it for my segmentation task.

---

> ### Author Response · Authors · 2022-11-16
> **Author response 1/4**
>
> **Dear Reviewer jgui,**
>
> we would like to thank you for your detailed review and take the opportunity to clarify and comment on the raised weaknesses in the order of the original review.
>
> ### Writing of the paper and definitions
>
> We appreciate this comment and have put substantial effort into improving the introduction, the Figures, and the method section. We  want to remark on our appreciation for this comment. Major rewriting, adaption, and restructuring have been done to:
>
> 1. The caption of Figure 1 has been updated. Here, the segmented foreground is white, and the background is black. The white foreground forms a total of 23 matched cycles. For presentation clarity, we only visualize six of the matched cycles in colors. Please do not compare the remaining white foreground as we did not show their corresponding cycles. In the Appendix, we added Figures 10, 11, and 12 which visualize randomly selected cycles and connected components for the other datasets. For clarification, we now reference Figure 1 in the introduction.
>
> 2. Moreover, we have adapted the captions of Figures 4,6 and rearranged the previous Figure 2 into two separate Figures (7 and 8) in the context of the loss and metric properties which they describe.
>
> 3. For better clarity, we have added the definitions of #ker and #coker to section 3.3.
>
> 4. We simplified Sections 3.1, 3.2, and 3.3 in regards to providing more details and improving understandability for the readers and moved Section 2.2, "Homology" to the Appendix.
>
>
> Furthermore, we have added minor improvements throughout the manuscript.
>
>
> ### Physical meaning of TopoMatch and definition of p:
>
> > 1) *"More explanation of the physical meaning of TopoMatch is desired."*
>
> We appreciate the pointed comment and take the opportunity to address the physical meaning of TopoMatch in a dedicated paragraph in section 3.2 of our manuscript. During training, we can interpret through its gradient direction, that matched topological features get emphasized, and unmatched features get suppressed. This property highlights the importance of finding a spatially correct matching because only then the correct features and, thereby, pixels will be emphasized, and the segmentation will be improved.
>
> During inference, TopoMatch serves as an interpretable error metric which is twice the number of wrong predictions, including false positive and false negative connected components (dim-0) and cycles (dim-1).
>
> Please note that we are not entirely sure if this entails the "physical meaning" the reviewer was referring to. In case we misinterpreted this review, we would kindly ask you to point this out so that we have the opportunity to address your concerns in time.
>
> > 2) *"What is p in the TopoMatch loss equation?"*
>
> In simple terms, **p** is a point in the persistence diagram.
> For clarification, **p is now changed** to q in our manuscript, which we explicitly define in section 3.2 in the beginning of the paragraph "Gradient of TopoMatch loss". Precisely: A point $q=(q_1,q_2) \in Dgm(L)$ describes a topological feature that is born by adding pixel $b(q)$ (**birth** of $q$) and killed by adding pixel $d(q)$ (**death** of $q$) to the filtration, see our new Figure 7.
> Further, we introduce a Figure 20 in the Appendix illustrating a persistence diagram and its relation to barcodes and their matching.
>
> ### Comparison image $C=max(L)$ and $C=min(L)$ and the use of two induced matchings
>
> > *"The comparison image is defined as C = min(L, G) in the main text, while in the caption of Fig.7, it is defined as C = max(L, G). Why the TopoMatch matching is the composition of two induced matching?"*
>
> We appreciate this comment and have added an entire Subsection 3.1 to our manuscript to clarify *Matching by comparison in ambient Space*. In simple terms, to compare $G$ and $L$, we need a **common ambient space**, which in our case is the **comparison image** $C$. Then, we match the features between $G$ and $C$; and between $L$ and $C$ by two separate matchings. Therefore, to guarantee a spatial correspondence between the features in $L$ and in $G$, we need the comparison image $C$ and the composition of the two **induced matchings**. Please note that in the sublevel setting, the comparison image $C$ has to be $C=max(L,G)$, and in the superlevel-setting it has to be $C=min(L,G)$, such that it creates a common ambient space; we apologize for the confusion in the manuscript and now consistently use the notion of sublevel filtrations $C=max(L,G)$ throughout our theory. For detail, please refer to Section 3.1.

---

> > ### Author Response · Authors · 2022-11-16
> > **Author response 2/4**
> >
> > ### Main contribution of the paper
> >
> > > *"The contribution of this paper should be explicitly pointed out. It seems like the main contribution is the use of two-induced matching to define a new metric, which seems to be incremental."*
> >
> > To describe our main contribution in simple terms, we would like to reiterate our general comment:
> >
> > *The spatial correspondence of topological features is a crucial property in segmentations of complex structures such as roads or neurons.
> > Preserving the complex shape and connectivity of structures such as vessels and neurons is a difficult task in segmentation.
> > Such structures can be described in terms of topological features and their spatial correspondences.
> > No existing method takes **spatial** correspondences of topological features into account.
> > To address this deficit, we introduce concepts from algebraic topology  (specifically, induced matchings of persistence barcodes) to machine learning.
> > Precisely, our method can guarantee that the connected components (Betti 0) and cycles (Betti 1) are matched in a spatially correct manner in the optimization function (see Figures 1 and 2). This guarantee of correct matchings sets us apart from the existing literature: we show that existing approaches lead to incorrect matchings in more than 99 percent of cases. Moreover, our TopoMatch construction experimentally proves to be an effective loss function and interpretable metric for image segmentation, leading to vastly improved topology across 6 diverse datasets.*
> >
> > **In one sentence: our central contribution is the first differentiable solution for localized topological error finding in Images.**
> >
> > We understand that the clarity had to be improved in the manuscript. Therefore we also extended our contribution section to describe these main contributions more explicitly. We hope these extensions lead to greater clarity and more fun when reading the manuscript.
> >
> > ### Experimentation -- Results and the Interpretation
> >
> > We appreciate the concrete comments on our experimentation and would like to provide replies to the specific comments in a point-by-point fashion:
> >
> >
> > > 1) *"Experimental results do not show impressive improvement or a good balance of segmentation accuracy and topology correctness"*
> >
> > To address this point, which is similar to reviewer GXoX, we now consider the standard deviation of all metrics across the test set. We now boldface the highest mean value in a dataset if it is at least $std/8$ away from the next best-performing method. We think that this way of boldfacing now better illustrates the magnitude of the differences in metrics. We show that the performance differences in volumetric metrics are extremely minor, whereas the performance of our method in topological metrics is consistently superior to all baselines. Note that naturally, the clDice Loss function optimizes the clDice metric.
> >
> > We would also like to reiterate the high importance of the topological metrics in the example of vessel segmentation, which has been widely discussed in the literature, e.g., Hu et al. and Shit et al. When segmenting vessels (or roads in satellite images), the most important property is their correct connectivity. This property is picked up in our topological metrics, whereas Dice and Accuracy purely focus on the total number of correctly segmented pixels. This will lead to a bias towards the largest vessels, e.g., big arteries, see Shit et al., Hu et al., Mosinska et al.
> >
> > Now, considering the quantitative and qualitative results, we conclude that our method *TopoMatch* achieves the right balance of segmentation accuracy and topology correctness.
> >
> > > 2) *"The experimental results show that the correction of the topology of segmentation hinders the improvement of segmentation accuracy."*
> >
> > For this comment, we would kindly reiterate on our previous comment and refer to Table 1. No method is consistently the best in volumetric scores, e.g. in Accuracy, each of the presented methods (including ours) is best in at least one dataset. Further, we did a statistical significance test on (Wilcoxon signed rank test; p-value < 0.01) for the Accuracy metric. The test revealed no statistically significant difference in the CREMI, synMnist, and Colon dataset. For the Buildings dataset our method is statistically significantly better than all baselines, and for the Roads dataset, the *clDice* metric is statistically significant compared to all other methods. Based on this analysis, we conclude that our method does not hinder the improvement of the segmentation Accuracy and, indeed, achieves very similar performance to the given baselines.
> >
> > Still, we very much understand that this messaging has not been entirely clear in our manuscript and have adapted Table 1, the caption, as well as the text in our manuscript.

---

> > > ### Author Response · Authors · 2022-11-16
> > > **Author response 3/4**
> > >
> > > > 3) *"Also, once the dice score is over 0.9, the topology correctness, even the spatial correspondences, would not be a big issue, as demonstrated by the Betti number."*
> > >
> > > We are not entirely sure if we correctly understand your comment; we anticipate that you are referring to Elegans and Colon, where the Dice is >0.9, and the T.M. metrics and the Betti error are lowest across the datasets. We would like to clarify that the topology metrics (T.M. and Betti) cannot be directly compared across datasets because they are physically interpretable quantities (# of connected components and # of cycles) that highly vary across the datasets and image sizes. For example, CREMI is evaluated on images of 312x312 pixels, which often include hundreds of cycles, and Elegans is evaluated on 96x96 pixels which often include few to no cycles. Clearly, the number of components and cycles influences the magnitude of the metric. Therefore, metrics can only be compared within the same dataset.
> > > Note that CREMI has a Dice close to 0.9 but has the highest overall topological errors.
> > >
> > >
> > > > 4) *"Is it necessary to have such a complex term in the loss function? Or can we choose a good segmentation mode and just use the Betti number as a constraint?"*
> > >
> > > Thank you for the remark; we would like to refer to Section 3.3 in our manuscript, which explicitly illustrates the shortcoming of the Betti error. In summary, the Betti number cannot consider the spatial location of cycles and connected components. Hence, any segmentation model which relies on or improves upon the Betti number will inherit its shortcomings and is topologically suboptimal.
> > >
> > > Regarding the second proposition, in order to incorporate *constraints* during inference, e.g., Betti number one would need prior information about the underlying number of cycles and connected components, which is not the case in challenging tasks such as roads or vessels or cells.
> > >
> > >
> > > > 5) *"It seems like improving the segmentation accuracy is another good way to correct the topological issues of segmentation. So, in what situations we do need the topology correctness term?"*
> > >
> > > Please refer to our previous comment 3 and our new Figure 2. Additionally, our table explicitly shows that despite being comparable in segmentation Accuracy, our TopoMatch vastly outperforms other methods in topological measure. This topological improvement is highly important for complex structured segmentation tasks, e.g., remote sensing data or medical images. For reference, please consider Henry et al. ICPR 2021, Can et al. CVPR 2022.
> > >
> > >
> > > Can, Yigit Baran, et al. "Topology Preserving Local Road Network Estimation from Single Onboard Camera Image." CVPR. 2022.*
> > >
> > > Henry, Corentin, et al. "Aerial Road Segmentation in the Presence of Topological Label Noise." ICPR, 2021.*
> > >
> > >
> > > ### Presentations of the meat of the paper
> > >
> > > > *"Perhaps more descriptions on the meat of this paper while concisely introducing some background and making connections to the method proposed in this paper."*
> > >
> > > Regarding this comment, we would like to refer to our general comment to all Reviewers as well as to our changes, which we describe in point **1.**. We made these in consideration of this comment and hope that our work now better conveys the connections between our theoretical background, the implemented methods, and our experimental results.
> > >
> > > ### Novelty:
> > >
> > > > *"Also, the novelty is limited .. "*
> > >
> > > To address the novelty concern, we would like to reiterate our main contributions.
> > >
> > > We know that the segmentation of the complex shapes and connectivity of structures such as vessels and neurons is a very difficult task. Therefore, the spatial correspondence of topological features is a crucial property in segmentations of complex structures such as roads or neurons because topological features can describe any real shapes. No existing method so far has taken *spatial* correspondences of topological features into account. Our method addresses this by introducing recently proposed concepts from algebraic topology  (specifically, induced matchings of persistence barcodes) to machine learning. In this, our method is entirely novel. For reference, the concept of induced matchings itself was only recently introduced to algebraic topology.
> > >
> > > In absence of any prior published work, which resembles our work's theoretical and experimental contributions, we believe that our work should be perceived as novel and non-incremental.
> > >
> > > Additionally, we would like to kindly point to the other reviewers who described the novelty of the paper as "non-trivial" and "quite original".

---

> > > > ### Author Response · Authors · 2022-11-16
> > > > **Author response 4/4**
> > > >
> > > >
> > > >
> > > > ### Reproducibility and code release:
> > > >
> > > > > *"... and the reproducibility of this work depends on the release of its source code."*
> > > >
> > > > We very much agree with the reviewer and share their view that the release of code is crucial to facilitate open science and reproducibility. We are happy to now release the **full code of our TopoMatch** algorithm, including the implemented baselines, evaluation, etc., in an anonymous GitHub repository. We would like to take this opportunity to point to a few resources in our repository that the Reviewers, as well as the community, may find useful. We include:
> > > >
> > > > 1. useful visualization and explanation notebooks that illustrate the difference and power of our TopoMatch compared to Wasserstein matching ('introduction.ipynb')
> > > >
> > > > 2. the full training routines and loss functions of all baselines. We think that this is a particularly useful contribution to the community of topology preserving image segmentation as currently, no comprehensive repositories exist which make the highly cited baseline methods of Hu et al. and Shit et al. available in a common and useable framework.
> > > >
> > > > 3. the full evaluation routine, which includes the calculation of a vast set of volumetric and topology-aware metrics to evaluate segmentation algorithms. This includes Dice, clDice, Accuracy, Voi, Ari, TopoMatch in dimensions 0 and 1, and Betti errors in dimensions 0 and 1.
> > > >
> > > > Finally, we would like to sincerely thank Reviewer jqui. We put in substantial effort to reorganize, clarify and rewrite the paper according to their input and believe that the paper is now in a way better shape to fit with the ICRL community.
> > > >
> > > > The authors

---

### Official Review · Reviewer_WAAt · 2022-10-24

**Confidence:** 4
**Correctness:** 3
**Technical Novelty And Significance:** 3
**Empirical Novelty And Significance:** 2
**Recommendation:** 6

**Clarity, Quality, Novelty And Reproducibility:**

Clarity could be improved with more toy examples. The proposed loss is quite original though.

**Strength And Weaknesses:**

Strengths:
---The proposed loss seems quite efficient from the numerical experiments
---It is rather original as topology-based segmentations are often based on standard distances between persistence diagrams

Weaknesses:
---I think the writing of the paper could be improved, as I find it quite difficult to follow for someone that is not aware of topological data analysis and its algebraic foundations.
---I think it would be interesting to compare against the simpler, more naive method of using the Wasserstein distances between L and min(L,G) and between min(L,G) and G. It seems that composing these two matchings would already provide better segmentations than the Wasserstein distance between L and G. Why are induced matchings better than this?
---Optimizing losses based on persistent homology usually requires heavy mathematics (http://proceedings.mlr.press/v139/carriere21a.html, https://link.springer.com/article/10.1007/s10208-021-09522-y), so I was wondering if the gradient of TopoMatch is well defined also, and if it enjoys some kind of convergence guarantees?

**Summary Of The Paper:**

In this paper, the authors propose a new loss and evaluation metric based on persistence theory for matching different image segmentations. The proposed loss can be quite different from the usual Wasserstein distance between persistence diagrams, and is rather based on the algebraic foundations of persistence theory, which turns out beneficial for practical purposes as the Wasserstein distances often induces matching errors. Finally the authors provide an exhaustive set of numerical experiments showcasing the efficiency of the new loss.

**Summary Of The Review:**

Overall, I think the paper is fine. The writing could be improved by adding some experiments from the supplementary inside the main body of the text and by simplifying a bit the theoretical section, but I think it is OK globally.

---

> ### Author Response · Authors · 2022-11-16
> **Author rebuttal**
>
> **Dear Reviewer WAAt,**
>
> We appreciate your positive comments on originality and novelty. We also understand your request for clarity and have put substantial effort into improving the introduction, the figures, and the method section in hope of providing greater clarity while maintaining the strong algebraic foundations of our work. We have put a specific focus on *simplifying the theoretical section 3*, where we adapted Sections 3.1, 3.2, and 3.3 in regards to providing more details and intuition for the readers. Second, we moved Section 2.2 "Homology" to the Appendix. Further, we restructured the previous Figure 2 and now present its content in two separate Figures (7 and 8) in the context of the loss and metric properties which they describe. We hope that these changes improve readability and clarity throughout.
>
> ### Composed Wasserstein Matching
>
> Importantly, we appreciate your excellent suggestion in regard to the **composed Wasserstein matching**. We implemented new experiments on the CREMI dataset and find that composed Wasserstein matching outperforms the previous state-of-the-art (Hu et al.) in topological metrics; see Table below. Still, our  **TopoMatch** performs substantially better than the suggested composed Wasserstein matching.
>
> Again, we really appreciate this pointed suggestion because this experiment depicts that the **composed** aspect of our work contributes, on one hand, but the **spatial aspect of the matching** is the leading reason for the superior performance of **TopoMatch**.
>
> | Loss             | Dice   | ClDice   | Accuracy | T.M.   | T.M.-0 | T.M.-1  | Betti    | Betti-0  | Betti-1 |
> |------------------|--------|----------|----------|--------|--------|---------|----------|----------|---------|
> | Dice             | 0.894  | 0.939    | 0.959    | 74.82  | 19.84  | 54.98   | 114.12   | 39.12    | 75.00   |
> | clDice           | 0.879  | 0.944    | 0.952    | 73.52  | 17.18  | 56.34   | 103.92   | 33.64    | 70.28   |
> | Hu et al.        | 0.888  | 0.935    | 0.957    | 81.24  | 22.12  | 59.12   | 118.16   | 43.68    | 74.48   |
> | **Composed W.M.**    | 0.895  | 0.941    | 0.959    | 72.95  | 20.32  | 52.63   |	106.64   | 40.00    | 66.64   |
> | **Ours (TopoMatch)** | 0.893  | 0.941    | 0.959    | **64.90**  | **15.50**  | **49.40**   | **79.16**    | **30.36**    | **48.80**   |
>
>
> From a theoretical standpoint, we would like to reply to the query, *"Why are induced matchings better than this? "*:
>
> The improvements of the composed Wasserstein matching over the direct Wasserstein matching (Hu et. al) can be explained by the diversity of intervals in the barcode of the comparison image compared to the barcode of the ground truth, which exclusively contains the interval $[0,1)$. Therefore, the length of the intervals in the barcode of the comparison image provides an indication of which intervals in the barcode of the likelihood map should be matched. Whereas, the direct Wasserstein matching simply matches bars by the length in descending order.
>
> The improvement of replacing the Wasserstein matching by the induced matchings comes from the fact that induced matchings do not consider the intensity of features as the matching criterion but their spatial correspondence within their respective images (see Fig. 1, 7). This leads to no false positive matches and no false negative matches. Following, this the gradient of our loss will always emphasize correct features and penalize features that do not exist in the ground truth (see Section 3.2 and Appendix. G).
>
>
> ### Well definition of the gradient and convergence guarantees
>
> We thank you again for pointing this out. We have explicitly added a Subsection in 3.2 in the main document on the topological gradients and now provide details on how the gradient is well defined in our setting. Additionally, we have provided a training loss plot that shows the convergence of TopoMatch in the appendix.
>
> In general, the complexity of considering the differentiability of functions defined on barcodes is simplified in our setting since the endpoints of intervals directly correspond to pixel-values within the corresponding image (see  Physical meaning of the gradient in Sec. 3.2). Hence, in fact, we are not considering a function defined by a barcode but rather a function defined on the image matrix.
>
> > *"adding some experiments from the supplementary inside the main body of the text*
>
> We appreciate this concrete suggestion for improving the clarity. We now add the ablation on the bothlevel vs. the superlevel matching to our main document. We hope this contributes to improving the writing and clarity.
>
> We want to thank reviewer WAAt for their review and time and kindly ask if there may be any other aspects that should be added or clarified.
>
> The authors

---

> > ### Comment · Reviewer_WAAt · 2022-12-05
> > **Response**
> >
> > Thank you for your clarifications and additional experiments. I am still a bit unsure about the general suitability of this work to ICLR but your answer allows for fruitful discussions among reviewers.

---

### Author Response · Authors · 2022-11-11
**General comment from the authors**

Dear Reviewers, dear Area Chair,

We highly appreciate the generally positive feedback on the practical and theoretical contributions of TopoMatch.
We thank you for the comments on the clarity of the manuscript, and we will make a sincere effort to improve the introduction, the figures, and the method section.

We would like to take the opportunity to summarize and discuss our main contributions in simpler terms in hope of igniting a fruitful and interactive rebuttal period.

The spatial correspondence of topological features is a crucial property in segmentations of complex structures such as roads or neurons.
Preserving the complex shape and connectivity of structures such as vessels and neurons is a difficult task in segmentation.
Such structures can be described in terms of topological features and their spatial correspondences.
No existing method takes *spatial* correspondences of topological features into account.
To address this deficit, we introduce concepts from algebraic topology  (specifically, induced matchings of persistence barcodes) to machine learning.
Precisely, our method can guarantee that the connected components (Betti 0) and cycles (Betti 1) are matched in a spatially correct manner in the optimization function (see Figures 1 and 2). This guarantee of correct matchings sets us apart from the existing literature: we show that existing approaches lead to incorrect matchings in more than 95 percent of cases. Moreover, our TopoMatch construction experimentally proves to be an effective loss function and interpretable metric for image segmentation, leading to vastly improved topology across 6 diverse datasets.

Finally, we are pleased to release all of our code in an **anonymous GitHub repository** (https://anonymous.4open.science/r/TopoMatch-ED20/README.md) to improve reproducibility.

[Update] We have now completed the individual reviews.

Conclusively we would like to provide a general note on the content and space limitations of the manuscript.
Because of the space limitations of 9 pages which will be extended to 10 in the case of acceptance, and because Reviewer WAAt and jgui suggested simplifying the theory, we moved Section 2.2 on Homology and the corresponding Figures to the Appendix. In case of acceptance, we will be able to move these to the main paper, which we believe will further improve the clarity and algebraic foundations in the paper.

With kind regards,

The authors

---

### Decision · Program_Chairs · 2023-01-20

**Decision:**

Reject

**Justification For Why Not Higher Score:**

While two reviewers recommend acceptance, their reviews are not overly positive, and the reviewers consistently point out the lack of clarity in particular.

**Justification For Why Not Lower Score:**

N/A

**Metareview: Summary, Strengths And Weaknesses:**

Summary:
The paper suggests a new loss function and evaluation metric for matching image segmentations based on persistent homology.

Strengths:
The loss is novel and interesting, and seems computationally advantageous.

Weaknesses:
- Clarity: The reviewers find the paper challenging to read.
- Experiments: Important baselines are missing.
- Suitability for ICLR: The reviewers are in doubt whether the work is right for the ICLR venue

The reviewers appreciate the novel approach, and with a bit of work to improve clarity and tailor the experiments to show why this approach is important for the ML crowd, this paper would be suitable for a top ML conference. But in its current state, the paper is unfortunately not ready.